# REPORT

# Cytoplasmic ribosomes on mitochondria alter the local membrane environment for protein import

Ya-Ting Chang[1], Benjamin A. Barad[1,2], Juliette Hamid[3], Hamidreza Rahmani[1], Brian M. Zid[3], and Danielle A. Grotjahn[1]

**Most of the mitochondria proteome is nuclear-encoded, synthesized by cytoplasmic ribosomes, and targeted to the mitochondria posttranslationally. However, a subset of mitochondrial-targeted proteins is imported co-translationally, although the molecular mechanisms governing this process remain unclear. We employ cellular cryo-electron tomography to visualize interactions between cytoplasmic ribosomes and mitochondria in *Saccharomyces cerevisiae*. We use surface morphometrics tools to identify a subset of ribosomes optimally oriented on mitochondrial membranes for protein import. This allows us to establish the first subtomogram average structure of a cytoplasmic ribosome at the mitochondrial surface in the native cellular context, which showed three distinct connections with the outer mitochondrial membrane surrounding the peptide exit tunnel. Further, this analysis demonstrated that cytoplasmic ribosomes primed for mitochondrial protein import cluster on the outer mitochondrial membrane at sites of local constrictions of the outer and inner mitochondrial membranes. Overall, our study reveals the architecture and the spatial organization of cytoplasmic ribosomes at the mitochondrial surface, providing a native cellular context to define the mechanisms that mediate efficient mitochondrial co-translational protein import.**

## Introduction

Mitochondria are essential double-membrane organelles required for energy production, metabolism, and stress signaling in eukaryotic cells. While mitochondria have their own genome and protein synthesis machinery, 99% of mitochondrial proteins are encoded by nuclear genes and synthesized by cytoplasmic ribosomes (Wiedemann and Pfanner, 2017; Pfanner et al., 2021). These proteins are targeted to mitochondria via the mitochondrial targeting sequence recognized by the receptors of the translocase of the outer membrane (TOM), which imports them across the outer mitochondrial membrane (OMM). Proteins destined for the inner mitochondrial membrane (IMM) or matrix are transported by translocases of the inner membrane (TIM) (Chacinska et al., 2009). Defects in these pathways result in neurodegenerative and metabolic diseases (MacKenzie and Payne, 2007), highlighting the importance of mitochondrial protein import for maintaining cellular health.

Most nuclear-encoded mitochondrial proteins are synthesized by cytoplasmic ribosomes and then imported posttranslationally into the mitochondria (Chacinska et al., 2009; Avendaño-Monsalve et al., 2020). However, early electron microscopy studies revealed cytoplasmic ribosomes near the OMM (Kellems et al., 1975),

suggesting that some proteins may be co-translationally imported. Later work also showed that a population of mRNAs encoding mitochondrial proteins was enriched near the mitochondria (Suissa and Schatz 1982; Marc et al., 2002; Saint-Georges et al., 2008; Tsuboi et al., 2020). Further, proximity-specific ribosome profiling identified a subset of mitochondrial proteins, highly enriched for IMM proteins, that are subject to co-translational import (Williams et al., 2014). The nascent polypeptide-associated complex (NAC) has been shown to stimulate mitochondrial protein import by promoting the interaction between ribosomes and mitochondria (George et al., 2002; Lesnik et al., 2014). OM14 on the OMM was identified as a receptor for ribosome–NAC complex during co-translational import (Lesnik et al., 2014). In addition, the interaction between TOM complex and nascent peptide appears crucial for ribosome recruitment on the OMM (Pfanner et al., 2004; Gold et al., 2014; Schulz et al., 2015; Wang et al., 2020). However, the function, regulation, and molecular composition of mitochondrial co-translational import in native cells remains to be defined.

Cryo-electron tomography (cryo-ET) produces detailed three-dimensional (3D) reconstructions of organelles and macromolecules in their native cellular environment. When combined

[1]Department of Integrative Structural and Computational Biology, The Scripps Research Institute, La Jolla, CA, USA;   [2]Department of Chemical Physiology and Biochemistry, School of Medicine, Oregon Health and Science University, Portland, OR, USA;   [3]Department of Chemistry and Biochemistry, University of California San Diego, La Jolla, CA, USA.

Correspondence to Danielle A. Grotjahn: grotjahn@scripps.edu.



with subtomogram averaging (STA), cryo-ET can visualize endogenous protein complexes at the subnanometer resolution (Young and Villa, 2023). While these methods have been used to study co-translation at the endoplasmic reticulum (ER) membrane (Pfeffer et al., 2012, 2015; Gemmer et al., 2023), mitochondrial co-translation is comparatively rare and more difficult to structurally characterize (Gold et al., 2017; de Teresa-Trueba et al., 2023). Therefore, the regulatory mechanisms that stabilize ribosome–mitochondria associations for co-translational import remain largely unknown.

We used cryo-focused ion beam (cryo-FIB) milling and cryo-ET to capture associations between cytoplasmic ribosomes and mitochondrial membranes in *Saccharomyces cerevisiae* (*S. cerevisiae*). We leveraged our previously developed surface morphometrics pipeline (Barad et al., 2023) to identify cytoplasmic ribosomes with their peptide exit tunnel optimally oriented for co-translational import on the OMM. STA analysis of this population revealed the first structure of mitochondria-associated ribosomes oriented for protein import and identified multiple contacts between the ribosome and OMM formed around the peptide exit tunnel. We show that these ribosomes cluster on the OMM in an arrangement suggestive of polysome formation. Surprisingly, we observed a decrease in the OMM–IMM distance locally at co-translational import sites, suggesting these membrane regions may be optimally remodeled for efficient protein import. Our study provides insight into the previously uncharacterized structural interactions of cytoplasmic ribosomes at mitochondrial membranes that facilitate mitochondrial protein import in cells.

## Results and discussion

### Cellular cryo-ET captures cytoplasmic ribosomes surrounding mitochondria in native cellular conditions

Given that mitochondrial protein co-translation events are relatively rare, we sought to optimize conditions for enriching mitochondrially-associated ribosomes in *S. cerevisiae*. Prior work showed increased mitochondrial mRNA localization under respiratory growth relative to fermentative growth (Tsuboi et al., 2020), suggesting mitochondrial protein co-translation may similarly increase under these conditions. Additionally, treatment with the translation elongation inhibitor cycloheximide (CHX) arrests ribosomes on the OMM and increases their co-purification with mitochondria (Gold et al., 2017). Therefore, we grew yeast in fermentative or respiratory conditions with or without CHX before vitrification (Fig. 1 A). We used cryo-fluorescence microscopy (cryo-FM) to screen grids and select targets based on quality (cell density and ice thickness) (Fig. 1 B), followed by cryo-FIB milling to generate ~150–200-nm lamella (Fig. 1 C) and tilt series collection (pixel size = 2.64 Å). We observed ribosomes at the surface of mitochondria in the reconstructed tomograms (Fig. 1 D and Fig. S1 A), with CHX treatment increasing ribosome associations in both growth conditions (Fig. S1 B). These findings confirmed that CHX enriches cytoplasmic ribosomes at mitochondrial membranes in cells, consistent with earlier in vitro work (Gold et al., 2017).

### Surface morphometrics pipeline identifies cytoplasmic ribosomes optimally positioned for protein import on mitochondrial membranes

We set out to define cytoplasmic ribosome–OMM interactions in their cellular context using STA. The resolution obtained in STA depends on factors like magnification (pixel size), signal-to-noise ratio, and the number of macromolecules (Young and Villa, 2023). Considering these, we collected datasets of CHX-treated cells at higher magnification (pixel size = 1.66 Å) (Fig. S1 C). We developed a membrane-guided approach to identify ribosomes optimally positioned for mitochondrial protein import (Fig. 1, E–H). Using automated 3D template matching software (Hrabe et al., 2012; Chaillet et al., 2023; Maurer et al., 2024), we identified 35,784 ribosomes from 91 tomograms (Fig. 1 E). We refined the positions and orientations to produce an 8 Å map of the 80S ribosome (Fig. 1 E and Fig. S1 D).

Next, we used the surface morphometrics pipeline (Barad et al., 2023) to generate surface mesh reconstructions of mitochondrial membranes from voxel segmentations (Fig. 1 F). The distance and orientation of individual ribosomes relative to the nearest OMM mesh triangle were calculated using Python scripts (Fig. 1 G), identifying 2,823 ribosomes within 0–250 Å of the OMM. Ribosomes engaged in mitochondrial protein import are not only near the OMM but likely also adopt an orientation similar to that of co-translating ribosomes on the ER (Pfeffer et al., 2015; Gold et al., 2017) and purified mitochondrial membranes (Gold et al., 2017), with the peptide exit tunnel positioned within 95 Å from the membrane (Fig. 1 H). Using this cutoff, we identified 1,076 ribosomes optimally oriented for import, representing ~38% of ribosomes near the OMM under CHX treatment (Fig. 1 I and Fig. S1 E). This analysis shows that most ribosomes near the OMM are not optimally oriented for import and may represent pre-engagement states before import or random localization.

### Multiple contacts form between the OMM and the subset of cytoplasmic ribosomes optimally positioned for protein import

Previous studies have shown multiple contacts between cytoplasmic ribosomes and the ER membrane facilitate co-translational import into the lumen (Becker et al., 2009; Pfeffer et al., 2012, 2015; Jomaa et al., 2022; Gemmer et al., 2023; Jaskolowski et al., 2023). We hypothesized that similar contact points might exist between ribosomes and the OMM to support protein import. We performed STA of ribosomes optimally positioned for mitochondrial co-translational import (Fig. 1 I and Fig. S1 E) under CHX treatment, which produced a 19 Å resolution map (Fig. 2 A and Fig. S2 A; and Video 1). This structure revealed three contact points surrounding the peptide exit tunnel (Fig. 2 A and Fig. S2 B). These contact points are absent in a structure obtained by averaging ribosomes within 250 Å of the OMM but with their peptide exit tunnel facing away, suggesting that these associations are specific for import-oriented ribosomes (Fig. S2 C). Together, these data show multiple contacts between the OMM and import-oriented ribosomes, suggesting these connections likely stabilize co-translational protein import to mitochondria.

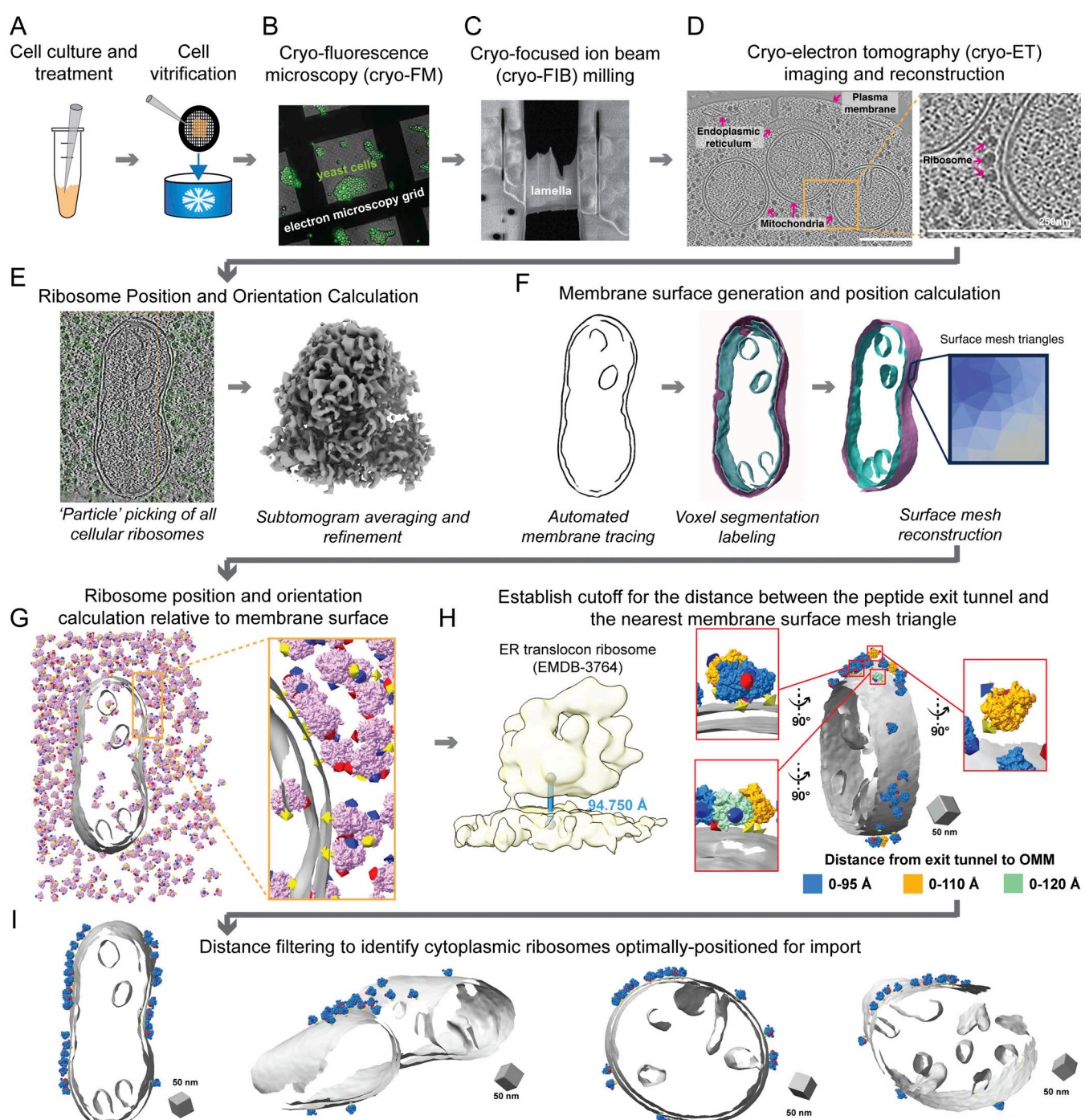

Figure 1. **Cellular cryo-ET imaging and processing workflow captures cytoplasmic ribosomes positioned for protein import on mitochondrial membranes. (A)** *S. cerevisiae* yeast expressing TIM50-GFP are grown in respiratory or fermentative conditions and treated with vehicle or CHX (100 µg/ml) prior to deposition on electron microscopy grids (black mesh circle) and vitrification via plunge freezing. **(B)** Vitrified yeast cells were imaged by cryo-FM to assess sample quality, cell density, and ice thickness. **(C)** Clumps of yeast were targeted for cryo-FIB milling to generate thin cellular sections (i.e., lamellae). **(D)** Cellular lamellae were imaged by standard cryo-ET acquisition procedures to generate tilt series that were further processed to generate 3D reconstructions (i.e., tomograms). Subcellular components such as mitochondria, the ER, the plasma membrane, and ribosomes are visible within the resulting tomograms. Scale bars = 250 nm. **(E)** Reconstructed tomograms were processed through "particle picking" software, which identified the initial positions and orientations of all visible cellular ribosomes. The positions and orientations were refined using subtomogram averaging to produce a consensus 8 Å 80S ribosome structure. **(F)** Mitochondrial membranes were traced, and separate three-dimensional voxel segmentations were generated for the OMM and IMM. These voxel segmentations were converted to surface mesh reconstructions using the surface morphometrics (Barad et al., 2023) pipeline such that the location of the membrane is represented by the coordinate of each triangle within the mesh. **(G)** The position and orientation of each ribosome relative to the OMM surface mesh reconstruction were calculated and rendered in the ArtiaX module of ChimeraX. The three-color arrows on ribosomes represent the Euler angles, with the yellow arrow representing the orientation of the ribosome peptide exit tunnel. **(H)** The cutoff for identifying cytoplasmic ribosomes engaged in protein import on the OMM was established by referring to the distance between the peptide exit tunnel of ER-translocon ribosome and the ER membrane. The

optimal cutoff of the distance between the exit tunnel and OMM was identified as 0–95 Å in ArtiaX as we started to observe the exit tunnel pointed away from OMM in the expanded cutoff, either 0–110 or 0–120 Å. **(I)** Cytoplasmic ribosomes optimally positioned for protein import were identified as those with their exit tunnel closer than 95 Å from the OMM.

To further characterize the contacts, we docked an atomic model of the 80S ribosome (PDB 4V6I) into the subtomogram average (Fig. 2 B). The expansion segment of the eS7La of the 25S rRNA in the large subunit corresponds to connection #1 (Fig. 2 B), similar to ER-associated ribosomes from purified microsomes (Gold et al., 2017) (Fig. S2 D). Despite its prevalence in both organellar import systems, the function of this connection remains unclear. Connection #2 forms immediately adjacent to the peptide exit tunnel, near ribosomal proteins rpL35, rpL26, and H5/6/7 from 5.8S rRNA (Fig. 2), which are the components associated with Ssh1 complex (Fig. S2 E) (Becker et al., 2009), the homolog of the Sec61 import channel in the ER. Given its proximity to the peptide exit tunnel, this connection may involve the nascent polypeptide chain and TOM complex, similar to the role of the Ssh1/Sec61 complex in mitochondrial protein import. Connection #3 forms between the ribosome and OMM near the eS27L expansion segment of 25S rRNA and rpL38 (Fig. 2). A similar interaction involving the rRNA expansion segment, rpL38, and the translocon-associated protein (TRAP) complex has been observed in ER-associated ribosomes, where it stabilizes cytoplasmic ribosomes on ER and ensures proper polypeptide targeting (Pfeffer et al., 2015; Gemmer et al., 2023; Jaskolowski et al., 2023) (Fig. S2 F). A predicted model of TOM20 has a loop-like structure resembling the rpL38A-interacting portion of the TRAP complex (Jaskolowski et al., 2023), suggesting it might be located near rpL38A to facilitate import.

Previous studies report a 33–35° difference in membrane orientation between ER- and mitochondria-associated ribosomes in *Schizosaccharomyces pombe* (EMD-14424, EMD-14423). In contrast, we observe no alignment difference between our structure and the ER-associated structure and only a slight 16° difference between ours and the mitochondria-associated structure from *S. pombe* (Fig. S2 G). These variations may stem from differences in ribosome orientation used for STA, as our method considers both distance and orientation relative to the OMM instead of distance alone. Additionally, these differences may reflect discrepancies in particle box size used for computational extraction, leading to additional membrane density in our structure relative to previous reports (de Teresa-Trueba et al., 2023). Finally, these structural differences may be CHX-dependent or species-specific, as no orientation differences were observed between the ER- and mitochondria-associated ribosome structures generated from purified mitochondria from CHX-treated *S. cerevisiae* (Gold et al., 2017).

In summary, our structure reveals three contacts between the cytoplasmic ribosome and the OMM in CHX-treated cells,

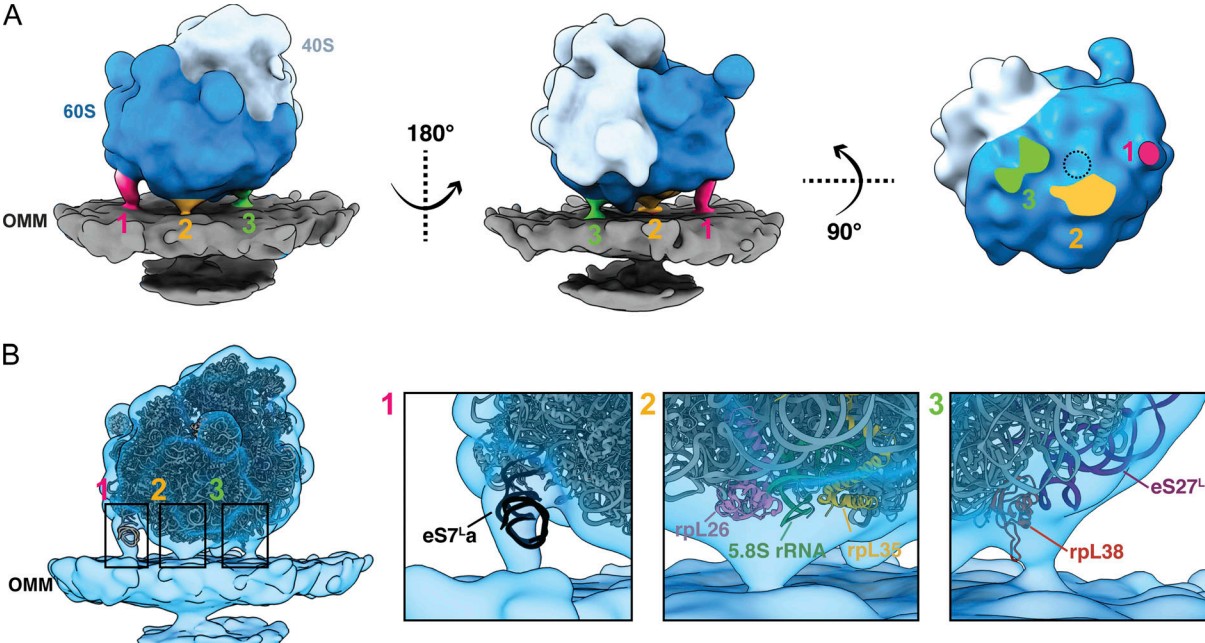

Figure 2. **3D subtomogram average of a cytoplasmic ribosome optimally positioned for protein import on the OMM shows multiple contact points.** **(A)** Three views of the subtomogram average of a cytoplasmic ribosome positioned with the exit tunnel on the 60S subunit (dark blue) facing the OMM (gray). Three connecting densities (labeled 1, 2, 3 in pink, orange, and green, respectively) are visible between the 60S and the OMM surrounding the peptide exit tunnel (dashed circle). **(B)** The subtomogram average of the mitochondria-associated ribosome (blue transparent density) with a fitted atomic model of the *S. cerevisiae* 80S ribosome (PDB 4V6I). Boxed regions focus on the cryo-EM density of each of the three connections observed between the cytoplasmic ribosome and the OMM.

resembling ER–ribosome interactions during co-translational import. These similarities enhance our understanding of the molecular components involved in mitochondrial co-translational import.

### Cytoplasmic ribosomes primed for protein import cluster on the mitochondrial membrane surface

Previous cryo-ET analysis on purified mitochondria from CHX-treated *S. cerevisiae* showed that mitochondrial-associated ribosomes cluster on the OMM (Gold et al., 2017). To assess whether similar clustering occurs in cells, we analyzed the spatial distribution of import-oriented ribosomes in our tomograms using Tomospatstat (Martin-Solana et al., 2024), which measures clustering based on the occurrence, $K(r)/K_{CSR}(r)$ (see Materials and methods). Our analysis showed that import-oriented ribosomes tend to cluster, with $K(r)/K_{CSR}(r)$ values above 1 (Fig. S3 A). Analysis of maximum $K(r)/K_{CSR}(r)$ values across 10-nm intervals showed greater clustering of import-oriented ribosomes on the OMM than non-import-oriented ones (Fig. 3 A and Fig. S3 B), particularly pronounced within 30–40 nm (Fig. 3 A). These results align with studies showing that 90% of import-oriented ribosomes cluster <50 nm apart on purified mitochondria (Gold et al., 2017).

Ribosome clusters can form stable polysomes that translate the same mRNA. Such polysome structures have been observed on the ER membrane both in vitro and within the native cellular context to facilitate co-translational import (Brandt et al., 2010; Pfeffer et al., 2012; Gemmer et al., 2023). Similarly, we observed import-oriented ribosome clusters on the OMM resembling polysomes, with mRNA entry and exit sites aligned (Fig. 3 B and Fig. S3 C). Fitting rigid rods between the 3′ mRNA entry and 5′ exit sites of adjacent ribosomes revealed putative mRNA pathways with end-to-end distances of ~223–689 Å, consistent with the estimated 5′ to 3′ lengths of mitochondrial-localized mRNAs associated with co-translational import (i.e., *TIM50*) (Guo et al., 2022). While this analysis suggests that these polysomes could accommodate a single mRNA, future work is needed to confirm whether clusters consist of single mRNAs with multiple ribosomes or independent mRNAs driving clustering.

Previous in vitro studies showed that import-oriented ribosomes preferentially associate with OMM regions near crista junctions (CJs) (Gold et al., 2017). To assess this in cells, we calculated OMM–IMM distances for OMM surface mesh triangles (Fig. 3, C and D). Visualization of distance measurements on the OMM surface meshes (Fig. 3 D) identified regions with the greatest OMM–IMM distances between CJs, consistent with prior findings (Barad et al., 2023). We used this feature to automatically partition "crista-associated" OMM patches (Fig. 3 F). Next, we identified the "co-translation ribosome-associated" patches and "non-co-translation ribosome-associated" patches on the OMM (Fig. 3 E) (See Materials and methods). We calculated and plotted the average overlap fraction between ribosome-associated OMM patches and crista-associated OMM patches (Fig. 3 F). We observed that import-oriented ribosomes showed significantly lower overlap compared with non-import-oriented ribosomes and random chance (Fig. 3 G).

This finding contrasts with previous reports showing import-oriented ribosomes cluster near CJs on purified mitochondria

(Gold et al., 2017). A possible explanation is that ribosomes near CJs may be more tightly associated and retained during purification, while those in non-CJ regions may destabilize. Regardless, our work highlights the value of contextual structural approaches for studying mitochondrial protein import in native cellular contexts.

Given the importance of ER–mitochondria contact sites in cellular functions (Csordás et al., 2018; Koch et al., 2024), we wondered whether mitochondrial co-translational import might be associated with these regions. Interestingly, we only observed one example of an ER membrane within 25 nm of the OMM in our data, with no overlap with co-translating ribosome-associated regions (Fig. S3 D). Collectively, these results show that import-oriented ribosomes cluster on the OMM but not near CJs nor ER–mitochondria contact sites, suggesting that alternate mechanisms may dictate ribosomal localization at the mitochondrial surface in the native cellular context.

### Cytoplasmic ribosome-associated protein import alters the local architecture of the outer and inner mitochondrial membranes

Previous work showed that most co-translationally imported proteins are IMM-targeted and require translocases on both the OMM and IMM for import (Williams et al., 2014). To explore if co-translational import is associated with local changes to mitochondrial membrane ultrastructure, we defined "co-translation-associated" and "non-co-translation-associated" OMM patches and measured the distance from each OMM triangle to the closest IMM triangle for these patches (Fig. 4 A). We observed significant reductions in the OMM–IMM distance in "co-translation-associated" patches relative to "non-co-translation-associated" patches, with most co-translating ribosomes located in regions where the OMM–IMM distance was <120 Å (Fig. 4, B and C; and Fig. S3, E and F). Despite fewer ribosomes optimally positioned for protein import in the vehicle-treated (i.e., non-CHX-treated) condition, a similar decrease in intermembrane distance was observed in co-translation-associated patches, suggesting that close contact is independent of translational arrest (Fig. 4 B).

The consistent decrease in intermembrane spacing in "co-translation-associated" regions suggests local membrane remodeling may facilitate efficient ribosome-mediated protein import. Notably, the OMM–IMM distances at "co-translation-associated" import sites align with previous quantum dot measurements of intermembrane spacing during TOM–TIM23-mediated protein import (Gold et al., 2014), suggesting the required OMM–IMM distance for import likely falls within this range. Previous cryo-ET studies of purified mitochondria found similar OMM–IMM distances in both protein-importing and non-importing areas (Gold et al., 2014). In contrast, we observed an average OMM–IMM distance of 130–140 Å in non-import sites, which is greater than in vitro measurements. This discrepancy may result from non-native membrane rearrangements during purification or limitations in prior quantification methods. Further, this highlights the importance of probing these functional interactions in the native cellular context.

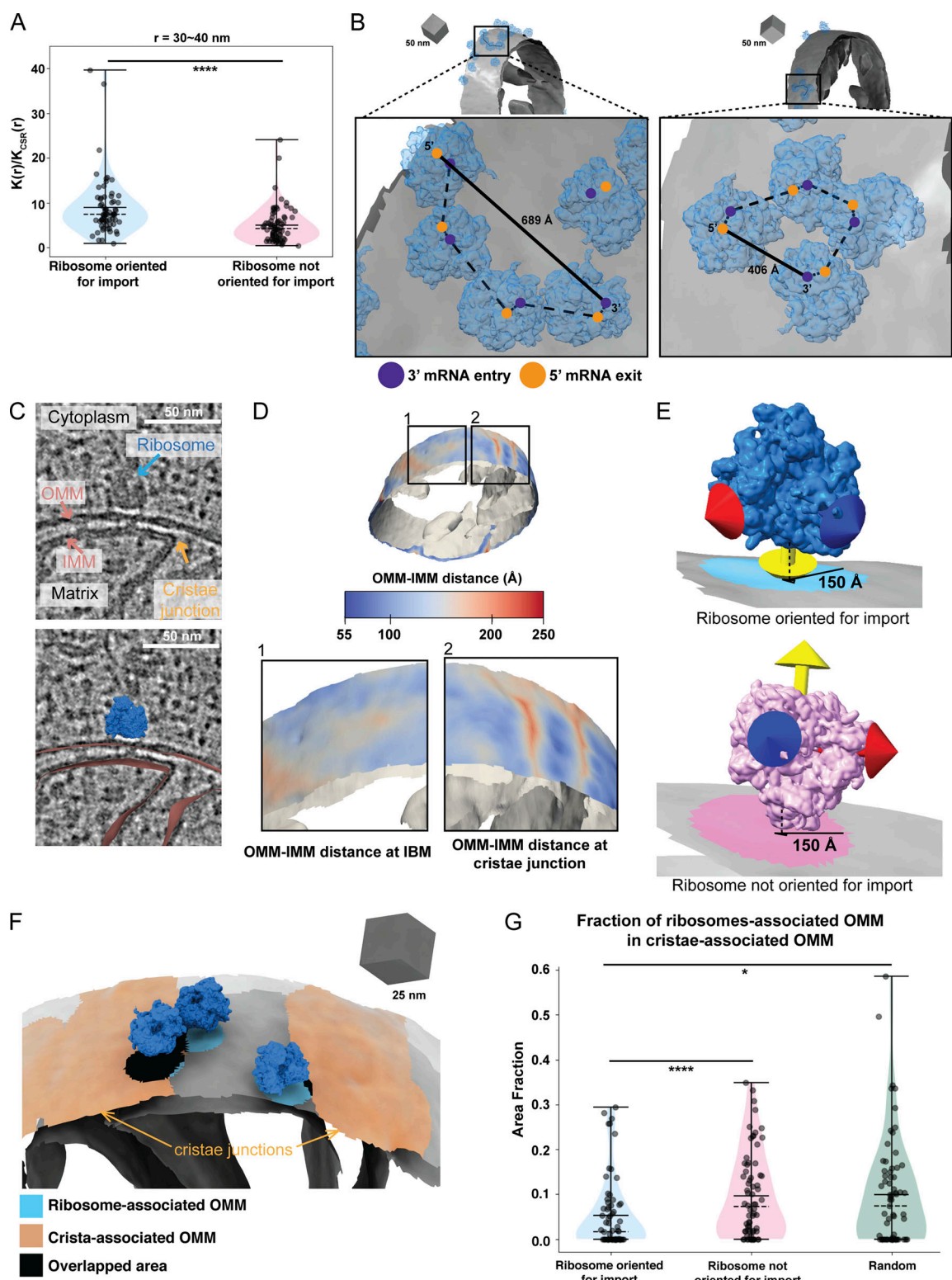

Figure 3. **Cytoplasmic ribosomes primed for protein import cluster on the mitochondrial membrane. (A)** Quantification of the maximum value of K(r)/K$_{CSR}$(r) for a 30–40-nm radius for each tomogram within the indicated ribosome class. Quantification from import-oriented ribosomes $n$ = 87 and non-import-oriented ribosomes $n$ = 89 tomograms are shown. P values from Mann–Whitney U test are indicated. ****P < 0.001. **(B)** Representative membrane surface reconstructions of mitochondria (gray) with ribosomes oriented for import relative to the OMM (blue). Insets show zoomed-in boxed regions of the ribosome models with circle overlays demarking the location of the 3' mRNA entry (blue), the 5' mRNA exit sites (orange), the possible pathways of interconnecting mRNA (dashed black line), and the calculated end-to-end distance from 5' to 3' of each interconnected mRNA (solid black line). **(C)** Representative tomogram slices showing labeled cytoplasm, ribosome, mitochondrial matrix, IMM, OMM, and CJs (upper panel) with an overlay of the surface mesh reconstructions of the IMM and OMM (red) and ribosome (blue) (lower panel). **(D)** Representative membrane surface reconstruction of mitochondria with the OMM surface colored

by OMM–IMM membrane distance and the IMM surface shown in gray. The bottom inset labeled "1" shows inner membrane boundary (IBM) regions on OMM with more subtle OMM–IMM distance variations. In contrast, the bottom inset labeled "2" shows regions on OMM with large OMM–IMM distances corresponding to CJs. **(E)** Ribosome and membrane models defining the patches on the membrane surface mesh reconstruction that correspond to ribosomes oriented for import (blue, top) and ribosomes oriented near but not oriented for import (pink, bottom). The ribosomes oriented for import are defined as those with the peptide exit tunnel (yellow arrow) pointed toward the membrane. In contrast, those not oriented for import have peptide exit tunnels facing away from the membrane. **(F)** Representative ribosome and membrane model with the OMM surface colored by the ribosome-associated (blue) and crista-associated OMM (orange), with areas of overlap (black). **(G)** Quantification of the average fraction of overlap from each tomogram between indicated ribosome class. Quantification from import-oriented ribosomes $n = 71$, non-import-oriented ribosomes $n = 71$, and random $n = 71$ tomograms are shown. P values from Mann–Whitney U test are indicated. *$P < 0.05$; ****$P < 0.001$.

## Conclusions

Recent advancements in cellular cryo-ET have enabled structural investigation of cellular proteomes in their native environment (Young and Villa, 2023). This work utilizes these developments to structurally characterize cytoplasmic ribosomes at the mitochondrial surface (Fig. 1). We show that models generated with our surface morphometrics approach (Barad et al., 2023) can identify macromolecules oriented relative to cellular membranes (Fig. 1, F–I and Fig. S1 E). We used this approach to identify, align, and average import-oriented cytoplasmic ribosomes (Figs. 2 and S2) from CHX-treated *S. cerevisiae*. To our knowledge, we present the first sub-tomogram average of a cytoplasmic ribosome forming three distinct connections with the OMM surrounding the peptide exit tunnel (Fig. 2). Using contextual morphometrics, we quantified local changes in membrane ultrastructure at import-associated regions, including OMM–IMM distance and clustering on the mitochondrial surface (Figs. 3, 4, and S3). Overall, this work sets the stage for enabling exciting opportunities to identify the molecular players regulating these interactions and local remodeling during mitochondrial protein import using genetic knockdown approaches.

## Materials and methods

### Yeast strains and growth conditions

The yeast strain used in this study is a derivative of the *S. cerevisiae* strain BY4741, which contains Su9-mCherry-Ura3 and TIM50-GFP-His3MX6. The TIM50-GFP strain was constructed by C-terminal tagging of the TIM50 open reading frame with GFP-HIS using the Pringle method (44836; Addgene) (Lee et al., 2013). They were recovered by streaking on the fresh YPD agar (1% yeast extract, 2% peptone, 2% glucose, and 2% agar) plates and incubated at 30°C for ~2 days. Yeast single colony on the plate was inoculated in YPD (1% yeast extract, 2% peptone, and 2% glucose) or YPG (1% yeast extract, 2% peptone, and 2% glycerol) and grown overnight at 30°C. The overnight culture was diluted to $OD_{600}$ of 0.2 with the corresponding medium and then grown to $OD_{600}$ of 0.8 at 30°C before vitrification. We verified that fluorescence labels did not impact mitochondrial health by performing a growth assay of cells from wild-type (BY4741) and fluorescently labeled strains (Su9-mCherry, TIM50-GFP). Cells were grown to the logarithmic phase in YPG media, and serial 10-fold dilutions of the same starting OD were then spotted (10 μl) onto YPG respiratory plates and incubated at 30°C (Fig. S1 F).

### Sample preparation for cryo-ET

Yeast liquid cultures with $OD_{600}$ of 0.8 was four times diluted to $OD_{600}$ of 0.2 with the medium supplemented with 133 μg/ml

CHX. The final concentration of CHX was 100 μg/ml. Yeast liquid cultures were incubated with CHX for 2 min, and then 4 μl of the sample was applied to the glow-discharged R1/4 Carbon 200-mesh gold EM grid (Quantifoil Micro Tools). The EM grid was incubated in the chamber of Vitrobot (Vitrobot Mark 4; Thermo Fisher Scientific) for another 2 min before it was plunge-frozen in a liquid ethane/propane mixture. The Vitrobot was set at 30°C with 100% humidity, and the blotting was performed manually from the back side of grids using Whatman #1 filter paper strips through the Vitrobot chamber side port.

### Cryo-FM for examining sample quality

Vitrified grids were clipped in Cryo-FIB Autogrid (Thermo Fisher Scientific) in the vapor of liquid nitrogen ($LN_2$). The clipped grids were loaded into the stage of Leica CryoCLEM microscope (Leica) to acquire the fluorescence/bright-field tiled image maps (atlases) in cryo condition by Leica LAS X software (25 μm Z stacks with system optimized steps, GFP channel ex: 470, em: 525). Z stacks were stitched together as a maximum projection map by LAS X navigator for examining the cell density and ice quality on EM grids. We also assessed mitochondrial network morphology using the fluorescence signal from Su9-mCherry to evaluate mitochondrial integrity.

### Cryo-FIB milling for lamella generation

Cryo-FIB milling of lamella was performed using Aquilos 2 cryo-FIB/SEM (Thermo Fisher Scientific) operated by software xT (Thermo Fisher Scientific). The atlases from cryo-FM were loaded into MAPS software (Thermo Fisher Scientific) to overlay with the scanning electron microscopy (SEM) atlases of the same grids. Before milling, EM grids were first subjected to a layer of platinum sputter for 15 s (1 kV, 20 mA, 10 Pa). Next, the grids were coated with an organometallic platinum layer using a gas injection system for 45 s and finally sputter-coated for 15 s (1 kV, 20 mA, 10 Pa). The regions of interest were selected in MAPS and then transferred to AutoTEM for identification of eucentric position, beam shifts, and tilt values in the preparation step. Before automated milling, the exact lamella positions were defined by the milling pattern with a width and height of 10 μm and thickness of 250 nm. After assigning the position of the lamella, the relief cuts were generated by a relatively high ion current (0.5 nA) with a width of 1 μm and a height of 6.5 μm. Next, the automated milling in AutoTEM was executed to remove bulk cellular material. The automated milling task was separated into three different steps. In the rough milling step, 0.5 nA ion current was applied to generate 2-μm thickness of lamella with 13-μm front width and 12-μm rear width. In the

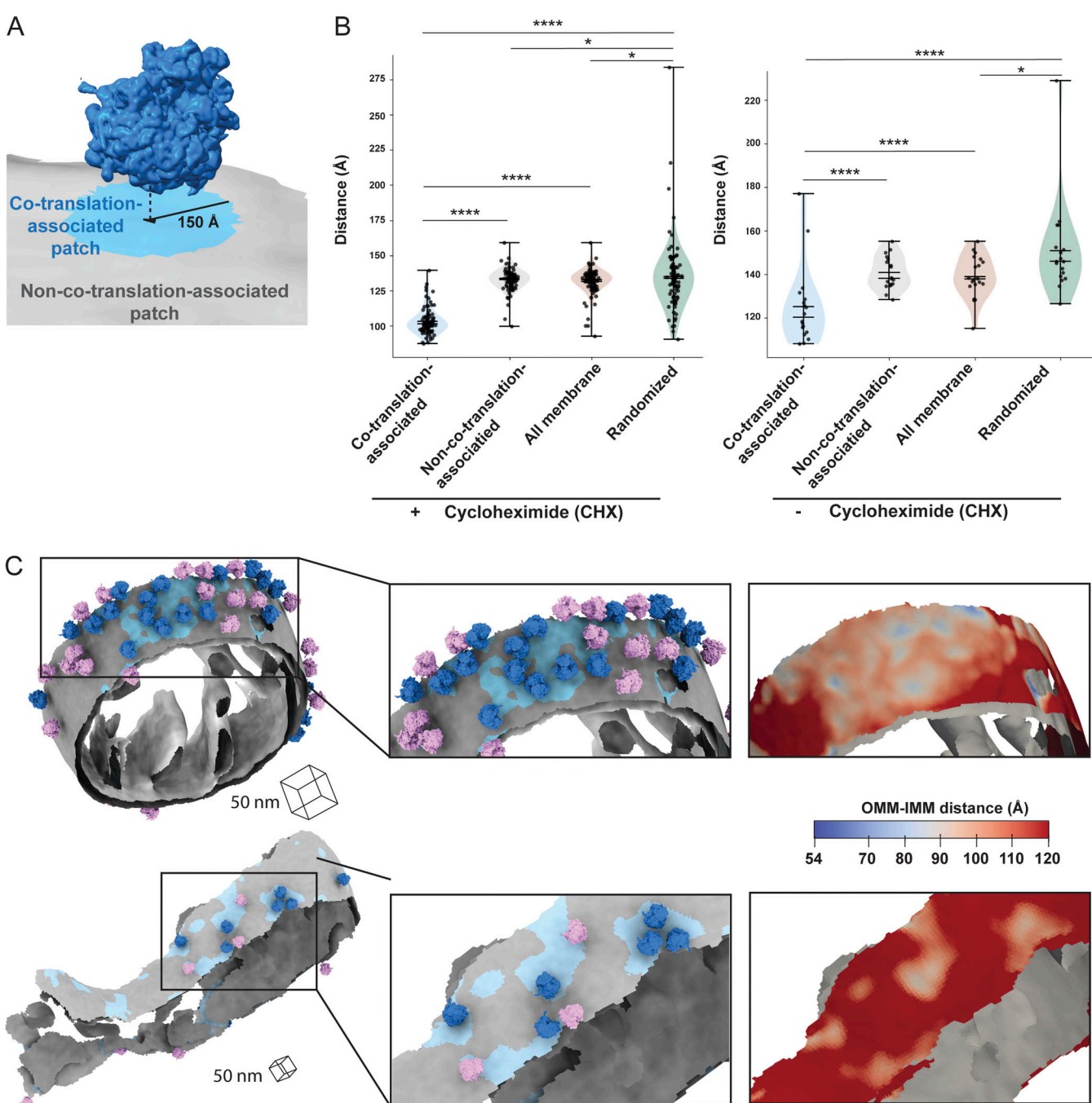

Figure 4. **Ribosome-associated protein import alters the local architecture of the outer and inner mitochondrial membranes. (A)** Ribosome and membrane model defining co-translation-associated and non-co-translation-associated patches on membrane surface mesh reconstruction for the OMM–IMM distance measurement. Co-translation-associated patches (blue) included the nearest OMM triangles (black in the middle of blue patch) to the import-oriented ribosomes and the OMM triangles within 150 Å of these nearest OMM triangles. Non-co-translation-associated patches (gray) consisted of the OMM mesh triangles that excluded co-translation-associated patches. **(B)** Quantification of the peak histogram values of OMM–IMM distance measurements for each tomogram within the indicated membrane patch region. The co-translation-associated and non-co-translation-associated patches are detailed in Fig. 4 A. The "all membrane" patches represent the entire OMM surface. The "randomized" patches were simulated according to the number of co-translation-associated patches, as outlined in Materials and methods. Quantification of CHX-treatment data $n$ = 87 and of vehicle-treatment data $n$ = 18 tomograms are shown. P values from Mann–Whitney U test are indicated. *P < 0.05; ****P < 0.001. **(C)** Representative membrane surface reconstruction of mitochondria colored by OMM–IMM membrane distance, with regions <10 nm shown in blue and regions >10 nm shown in gray. Ribosomes oriented for import relative to the OMM are colored blue, and the remaining ribosomes near but not oriented for import are shown in pink. Insets show zoomed-in boxed regions of the models (middle) and the local variations in OMM–IMM distance (right).

medium milling step, the 0.1 nA ion currents were applied to generate 1.2-µm thickness of lamella with 11.3-µm front width and 11-µm rear width. In the fine milling step, 50 pA was applied to generate a 600-nm thickness of lamella with 10.1-µm width. After the milling steps, the thinning step was automatically executed with an ion beam of 50 pA to generate 300-nm thickness of lamella with 10-µm width. A second thinning step was used to generate the 250-nm thickness of lamella with 10-µm width using a 50 pA ion beam. After the automatic milling and thinning process, a polishing step was manually executed using an ion beam of 50 pA and targeted for the thickness of lamella under 200 nm.

### Tilt series collection
Tilt series collection on the lamella was performed on Titan Krios (Thermo Fisher Scientific) operated at 300 keV and equipped with a K3 direct electron camera and a BioQuantum energy filter (Gatan). Individual lamellae were montaged with low dose (1 e/$Å^2$) at high magnification to localize cellular features and identify mitochondria by their distinctive OMM and IMM. Target selection and data acquisition were performed by Parallel cryo-ET (PACE-tomo 1.5) (Eisenstein et al., 2023), which is a set of Python-based SerialEM scripts allowing multiple tilt series collection in parallel on the same lamellae via beam shift. Data were acquired at a magnification of 53,000× with a pixel size of 1.6626 Å or 33,000× with a pixel size of 2.638 Å and a nominal defocus range between −4 and −6 µm with a 1-µm step. Tilt series collection was done in a dose-symmetric scheme with a 3° tilt increment and angles ranging from −60° and +60° centered on −11° pretilt with +11° starttilt. Data were collected with dose fractionation, with 10~11 0.28–0.33 e/$Å^2$ frames collected for each micrograph. The total dose per tilt is ~3 e/$Å^2$ and the total accumulated dose for the tilt series was ~120 e/$Å^2$.

### Tilt series processing and reconstruction
Motion correction and contrast transfer function (CTF) estimation were performed in Warp (Tegunov and Cramer, 2019). Tilt series stacks were generated in Warp after adjusting the starting tilt to 0° in the mdoc files using a python script (pretilt_mdoc.py). Tilt series alignment was performed using patch-tracking in batch mode and further optimized manually in etomo (Mastronarde, 1997). The resulting alignment files from etomo were imported into Warp. The resulting tomostar files from Warp were dose-adjusted by "adjust_tomostar.py." Tomograms were then reconstructioed with six times binning in Warp. This resulted in a voxel size of (9.98 Å)³ and dimensions of 682 × 960 × 334. Python scripts included in this process are available at https://github.com/hamid13r/warp_lamella_adapters.

### Template matching for localizing ribosome particles in tomograms
Cytoplasmic ribosome particles were localized by template-matching reconstructed tomograms against a yeast 80S ribosome (EMD-11096) by PyTom (Chaillet et al., 2023; Maurer et al., 2024). The template was modulated with the CTF function parameters according to the defocus of each tomogram at 300 keV voltage, 2.7 mm Cs, and 0.1 amplitude contrast. Then the templates were low-pass-filtered to 25 Å followed by downsampling to a pixel size of 9.98 Å. All template modulation was performed by "pytom_create_template.py" script. A spherical mask with a radius of 22-pixel size with a soft edge was created later by the "pytom_create_mask.py" script. Template matching was performed using an angular search of 7° using the "pytom_match_template.py" script. Low- and high-pass filters of 25 and 500 Å were, respectively, applied to templates and tomograms. Initial particle candidates were extracted by "pytom_extract_candidates.py" script with the maximum number of particles of 2,000. The optimized parameters with a 22-voxel masking radius and 10 number-of-false-positive were applied to perform the final extraction of ribosome particles as .star files.

### Subtomogram averaging analysis for cytoplasmic ribosomes
The annotation of cytoplasmic ribosome particles (.star files) was imported into Warp for subtomogram extraction, binned by 3 (pixel size of 4.99 Å). The subtomograms, CTF volumes, and the star file were transferred to Relion4 for STA analysis (Zivanov et al., 2022). We used relion_reconstruct to create the initial reference using the initial orientations from pyTOM and then refined all the particles using the auto-refine process. The aligned particles in Relion were further refined in M (Tegunov et al., 2021), resulting in a resolution of 10.3 Å (~Nyquist). The subtomograms aligned with M were extracted using 2× binning (pixel size = 3.33 Å). The 2× binned subtomograms from M were processed by the same Relion-M workflow as above. The final resolution of the cytoplasmic ribosome was determined to be 8 Å by the Fourier Shell Correlation (FSC) score of 0.143.

### Membrane tracing, voxel segmentation, and surface generation
The reconstructed tomograms with a pixel size of 9.98 Å were processed by Membrain-Seg (Lamm et al., 2024, *Preprint*), an advanced machine-learning software based on UNet for tracing and segmenting cellular membranes. The traced volumes of membranes were imported into AMIRA (Thermo Fisher Scientific) for manual curation. OMM and IMM were designated as different labels by 3D Magic Wand, and the ambiguous connection between them was manually curated slice-by-slice using the 2D Brush tool. Next, the individual labels of membrane voxel segmentation were reconstructed as smooth surfaces using the "segmentation_to_meshes.py" script with Angstrom units in surface morphometrics (Barad et al., 2023). The surfaces were generated with a maximum of 150,000 triangles, a reconstruction depth of 8, an interpolation weight of 8, and an extrapolation distance of 25 Å. The surface orientations were further refined using the "run_pycurv.py" script in surface morphometrics.

### Calculation of the relative distance and orientation between ribosomes and OMM surfaces and the relative distance between ribosome peptide exit tunnels and OMM surfaces
We used the relative distance and the angle to represent the relative localization and orientation between ribosomes and the nearest surface triangles. In the final refinement run, the starfile provided the coordinates (rlnCoordinateX, rlnCoordinateY, and rlnCoordinateZ) and orientation as Euler angles (rlnAngleRot,

rlnAngleTilt, and rlnAnglePsi) of cytoplasmic ribosomes. Additionally, the surface morphometrics pipeline provided the coordinates and orientation as normal vectors of surface triangles in triangle graph files (.gt). To determine the relative distance between ribosomes and OMM surfaces, we incorporated the coordinates from these data and utilized a k-dimensional tree Python function to calculate the minimum distance from each ribosome to the nearest OMM surface triangle. To calculate the relative angle between ribosomes and the nearest OMM surface triangles, we first converted the Euler angles of ribosomes to rotation matrixes through a Python function "euler2matrix" from the eulerangles package (https://github.com/alisterburt/eulerangles) and then extracted the vectors which visually represented as the three-color arrows in ArtiaX (Ermel et al., 2022), a plugin in ChimeraX (Pettersen et al., 2021). The relative angle between each vector of the ribosome and the normal vector of the nearest OMM surface triangle was calculated as the following equation:

$$\text{relative angle} = \text{across}(|\text{ribosome vector} \bullet \text{surface normal vector}|)$$

We positioned a spherical mask that covers the peptide exit of the 8 Å yeast 80S ribosome structure in ChimeraX and then determined the center of the mask in 3dmod (Kremer et al., 1996). We then shifted the center of the ribosome to the center of the peptide exit for all ribosome particles and extracted the particles to get the starfile with shifted coordinates using M (Tegunov et al., 2021). The relative distance between the peptide exit of ribosomes and the nearest OMM surface triangles was determined by the k-dimensional tree in Python as well.

The relative distance between ribosomes and OMM surfaces, the three relative angles between ribosomes and OMM surfaces, the identifiers of the nearest OMM surface triangles, and the relative distance between the peptide exit of ribosomes and OMM surfaces were recorded in the CSV files per tomogram for further analysis. This calculation is accomplished by "ribo_membrane_distance_orientation.py" script. The CSV files were further converted to starfiles per tomogram by the "csv_to_star.py" script for rendering particles in ArtiaX. The optimal cutoff for identifying the ribosomes oriented for protein import on mitochondria was determined visually using ArtiaX with a final threshold of 95 Å for the distance between peptide exit and OMM.

## Subtomogram averaging analysis for cytoplasmic ribosomes optimally oriented for protein import on mitochondrial surfaces

The starfile was filtered based on a final threshold of 95 Å for the distance between the peptide exit tunnel and the OMM to select the ribosomes optimally oriented for protein import into mitochondria. The orientation of selected ribosomes was visually examined in ArtiaX. This filtered and curated starfile was then input into Relion4 for initial alignment and averaging using Relion reconstruction (relion_reconstruct_mpi). The initial model was subsequently used as a reference for alignment and refinement in Relion auto-refine (relion_refine_mpi), resulting in a resolution of 19 Å. 30-Å low-pass filters were applied to the output model from Relion auto-refine using relion_image_handler to more clearly visualize the connection between the cytoplasmic ribosome and the OMM. The *S. cerevisiae* 80S ribosome (PDB 4V6I) was fitted into the 30-Å low-pass filtered map to study the ribosomal components within the map. Finally, the model from Relion auto-refine was further postprocessed in M to estimate the final resolution, determined by a FSC score of 0.143.

## Subtomogram averaging analysis for cytoplasmic ribosomes that are near but are not optimal for protein import on mitochondria

The starfile was filtered based on the following criteria to select the ribosomes that are in proximity to mitochondria but are not oriented for protein import: (1) the distance between ribosomes and OMM is less and equal to 250 Å, and (2) the coordinates are not included in the population of ribosomes that oriented for import. The filtered starfile was input into Relion for reconstruction (relion_reconstruct_mpi). The reconstructed model was used as a reference in Relion auro-refine (relion_refine_mpi).

## Analysis of spatial clustering of mitochondria-associated cytoplasmic ribosomes

We examined the clustering patterns of mitochondria-associated cytoplasmic ribosomes using Tomospatstat (Martin-Solana et al., 2024), which employs Ripley's K function K(r) to describe the occurrences of objects within certain distances r. For our analysis, the occurrence of the mitochondrial-associated cytoplasmic ribosomes on OMM in each tomogram within a given radius (r) is defined by K(r), and the occurrence that would be expected from complete spatial randomness (CSR) is defined by ($K_{CSR}(r)$). The $K(r)/K_{CSR}(r)$ ratio was further calculated to represent the level of clustering within r. We calculated this ratio for mitochondrial-associated cytoplasmic ribosomes oriented for protein import and for cytoplasmic ribosomes 250 Å away from mitochondria but not oriented for protein import, within r values ranging from 27 to 166 nm. The parameters we used in Tomospatstat were as follows:

$$\text{tomospatstat} - \text{K} - \text{k } 332 - \text{p } 0.499$$

The $K(r)/K_{CSR}(r)$ ratios along r for these two ribosome populations were recorded as CSV files for each tomogram. The curves of $K(r)/K_{CSR}(r)$ ratios along r were plotted using Matplotlib. The maximum values of $K(r)/K_{CSR}(r)$ for given radius intervals of 10 nm for each tomogram were visualized using violin plots. The Mann–Whitney U test was applied to assess the statistical significance of differences in the maximum values of $K(r)/K_{CSR}(r)$ between the two ribosome populations. The generation of violin plots and statistical tests were performed by "plotting_tomospatstat_K_ratio_stats.py."

## Identifying polysomes on mitochondrial membranes

We identified the positions of the 5′ mRNA exit and 3′ mRNA entry in our 8-Å cytoplasmic ribosome map by fitting the model of *S. cerevisiae* 80S ribosome (PDB 4V6I) in ChimeraX. Using the "Map Eraser" tool in ChimeraX, we cropped a sphere with a 20-Å radius from the map of the cytoplasmic ribosome at the 5′ mRNA and 3′ mRNA positions, saving these sphere maps to mark the locations of the 5′ mRNA exit and 3′ mRNA entry.

To display the optimally oriented cytoplasmic ribosomes along with their 5′ mRNA exit and 3′ mRNA entry on the mitochondrial

surface, we loaded the surface files (.stl) of the OMM and IMM, and the starfile of optimally oriented cytoplasmic ribosomes rendered with the maps of the cytoplasmic ribosome, 5′ mRNA exit, and 3′ mRNA entry in ArtiaX. We then manually searched for potential polysomes using the following criteria: (1) the nearest neighboring ribosome is within 30 nm, and (2) the 3′ mRNA entry is directly adjacent to the 5′ mRNA exit of the nearest neighboring ribosome. To show the interconnecting pathway of mRNA associated with polysomes, we placed markers at the 5′ mRNA exit and 3′ mRNA entry for each ribosome and connected the markers with dashed rods in the order of mRNA from 5′ to 3′ using the distance measurement tool in ChimeraX. We measured the end-to-end distance from 5′ to 3′ of these polysome-associated mRNAs using the distance measurement tool in ChimeraX.

### Analysis of the overlap fraction of "ribosome-associated OMM" in "crista-associated OMM"

We identified the ribosome-associated OMM regions for cytoplasmic ribosomes oriented for protein import and for those 250 Å away from the OMM but not oriented for protein import. To pinpoint these OMM regions, we first extracted the identifiers of the nearest OMM surface triangles for these ribosomes using the "match_particles.py" script. We then used these identifiers to locate the coordinates of the nearest OMM surface triangles from the OMM triangle graph files (.gt) and searched for triangles within a radius of 150 Å around these nearest OMM surface triangles using "OMM-patches_IMM_dist_measurement.py." We saved two separate OMM triangle graph files (.gt) with the label as ribosome-associated OMM for these two populations of ribosomes. Randomized ribosome-associated OMM regions were generated for each tomogram by "random_patches_OMM-IMM_dist_measurement.py" with the criteria as follows: (1) the number of randomized OMM regions matched the number of ribosome-associated OMM regions that associated with the cytoplasmic ribosomes oriented for import, and (2) the distances between the centers of the randomized regions were >150 Å. The separate OMM triangle graph files (.gt) with the label as ribosome-associated OMM were also saved for randomization.

To identify crista-associated OMM, we first subclassified CJs by measuring the distance from the IMM to the OMM using surface morphometrics with the "measure_distances_orientations.py" script. We selected the IMM triangles that were 18–30 nm away from the OMM and manually cleaned up the selected IMM triangles that did not belong to CJs. We extracted the subclassified IMM with manual curation as CSV files, which contained the identifiers of the nearest OMM surface triangles in Paraview. We then added a new label as CJ projected OMM for the OMM triangles with these identifiers in OMM triangle graph files (.gt) using "CJ_projected_OMM.py." Next, we searched the OMM triangles within 15 nm (half the width of a cristae body) of the CJ projected OMM and added a new label as crista-associated OMM for these OMM triangles in the OMM triangle graph files (.gt) using "expand_CJ_projected_OMM.py."

We identified the OMM triangles with both labels of "ribosome-associated OMM" and "crista-associated OMM" and added a new label for those triangles as "overlap_with_CJ_projected_OMM"

in the OMM triangle graph files (.gt) by "find_overlapping_trianlges.py." To quantify the overlap fraction of ribosome-associated OMM in crista-associated OMM per tomogram, we first counted the area of the OMM triangles with the label of ribosome-associated OMM, and the area of the OMM triangles with both labels of ribosome-associated OMM and crista-associated OMM from OMM triangle graph files (.gt). We then calculated the overlap fraction as follows by "count_overlap_area.py":

$$
\begin{aligned}
\text{Overlap fraction} = & (\text{Area of the OMM labeled with "ribosome} \\
& - \text{associated OMM" and "crista} \\
& - \text{associated OMM"}) / \\
& (\text{Area of the OMM labeled with "ribosome} \\
& - \text{associated OMM"})
\end{aligned}
$$

The overlap fraction per tomogram for both populations of ribosomes and the randomization was plotted as a violin plot. The Mann–Whitney U test was applied to assess the statistical significance of differences in the overlap fraction. The generation of the violin plot and statistic test were performed using "plotting_overlap_fraction_stats.py."

### Calculation of distances between OMM and IMM at co-translation-associated and non-co-translation-associated patches on the OMM

We defined co-translation-associated patches as areas where ribosomes oriented for import are associated with the OMM surface. To identify these patches, we first extracted the identifiers of the nearest OMM surface triangles for these ribosomes using the "match_particles.py" script from the CSV files by putting the starfile. We then used these identifiers to locate the coordinates of the nearest OMM surface triangles from the OMM triangle graph files (.gt) and searched for triangles within a radius of 150 Å around these nearest OMM surface triangles as ribosome-associated patches. The distances between ribosome-associated patches and IMM were further calculated using the k-dimensional tree in Python. The identification of co-translation-associated patches and the distance calculation were accomplished by "OMM-patches_IMM_dist_measurement.py." On the other hand, "outside_OMM-patches_IMM_dist_measurement.py" was applied to extract non-co-translation-associated patches by excluding co-translation-associated patches and to calculate the distances between non-co-translation-associated patches on the OMM and IMM. Randomized co-translation-associated patches were generated for each tomogram by "random_patches_OMM-IMM_dist_measurement.py" based on the following criteria: (1) the number of randomized patches matched the number of co-translation-associated patches, and (2) the distances between the centers of the randomized patches were >150 Å. The distance calculation between the complete OMM and IMM was performed by surface morphometrics using "measure_distances_orientations.py." Violin plots were created by generating histograms with 100 bins for each tomogram and identifying the peak value of the most populated bins. The Mann–Whitney U test was used to analyze the statistically significant differences in peak positions. The generation of violin plots and the statistical test were accomplished by "plotting_OMM-patches_IMM_dist_stat.py."

## Online supplemental material

Fig. S1 shows representative tomograms of cryo-FIB tomograms milled *S. cerevisiae* cell lamellae with visible mitochondria-associated cytoplasmic ribosomes. Fig. S2 shows a 3D subtomogram average of a cytoplasmic ribosome positioned for protein import on the OMM. Fig. S3 shows cytoplasmic ribosomes primed for protein import cluster on the mitochondrial membrane. Video 1 shows a 3D subtomogram average of a cytoplasmic ribosome optimally positioned for protein import on the OMM.

## Data availability

All tilt series, reconstructed tomograms, voxel segmentations, particle positions, and reconstructed mesh surfaces used for quantifications were deposited in the Electron Microscopy Public Image Archive (EMPIAR) (Iudin et al., 2023) under accession codes EMPIAR-12534. The final maps resulting from subtomogram were deposited in the Electron Microscopy Data Bank under accession codes EMD-48751 and EMD-48752. All scripts generated for these analyses are available at https://github.com/GrotjahnLab/co-translating_ribosome_scripts). Assistance from ChatGPT-3.5 (Open AI, https://chat.openai.com/) and Grammerly's AI tool was utilized to improve the clarity, grammar, and conciseness of the manuscript text.

## Acknowledgments

We thank Bill Anderson and William Lessin for microscope support and Jean-Christophe Ducom for computational support. We also thank R. Luke Wiseman for his critical input on the manuscript.

D.A. Grotjahn is supported by Nadia's Gift Foundation Innovator Award of the Damon Runyon Cancer Foundation (DRR-65-21) and the National Institutes of Health (NIH) grant RF1NS125674. B.M. Zid is supported by the NIH grant R35GM128798. This work used equipment supported by NIH grant S10OD032467.

Author contributions: Y.-T. Chang: Conceptualization, Data curation, Formal analysis, Investigation, Methodology, Software, Validation, Visualization, Writing - original draft, Writing - review & editing, B.A. Barad: Formal analysis, Methodology, Software, Validation, Writing - review & editing, J. Hamid: Investigation, H. Rahmani: Methodology, Software, Validation, Writing - review & editing, B.M. Zid: Conceptualization, Funding acquisition, Resources, Writing - review & editing, D.A. Grotjahn: Conceptualization, Funding acquisition, Project administration, Supervision, Writing - original draft, Writing - review & editing.

Disclosures: The authors declare no competing interests exist.

Submitted: 17 July 2024

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

# Supplemental material

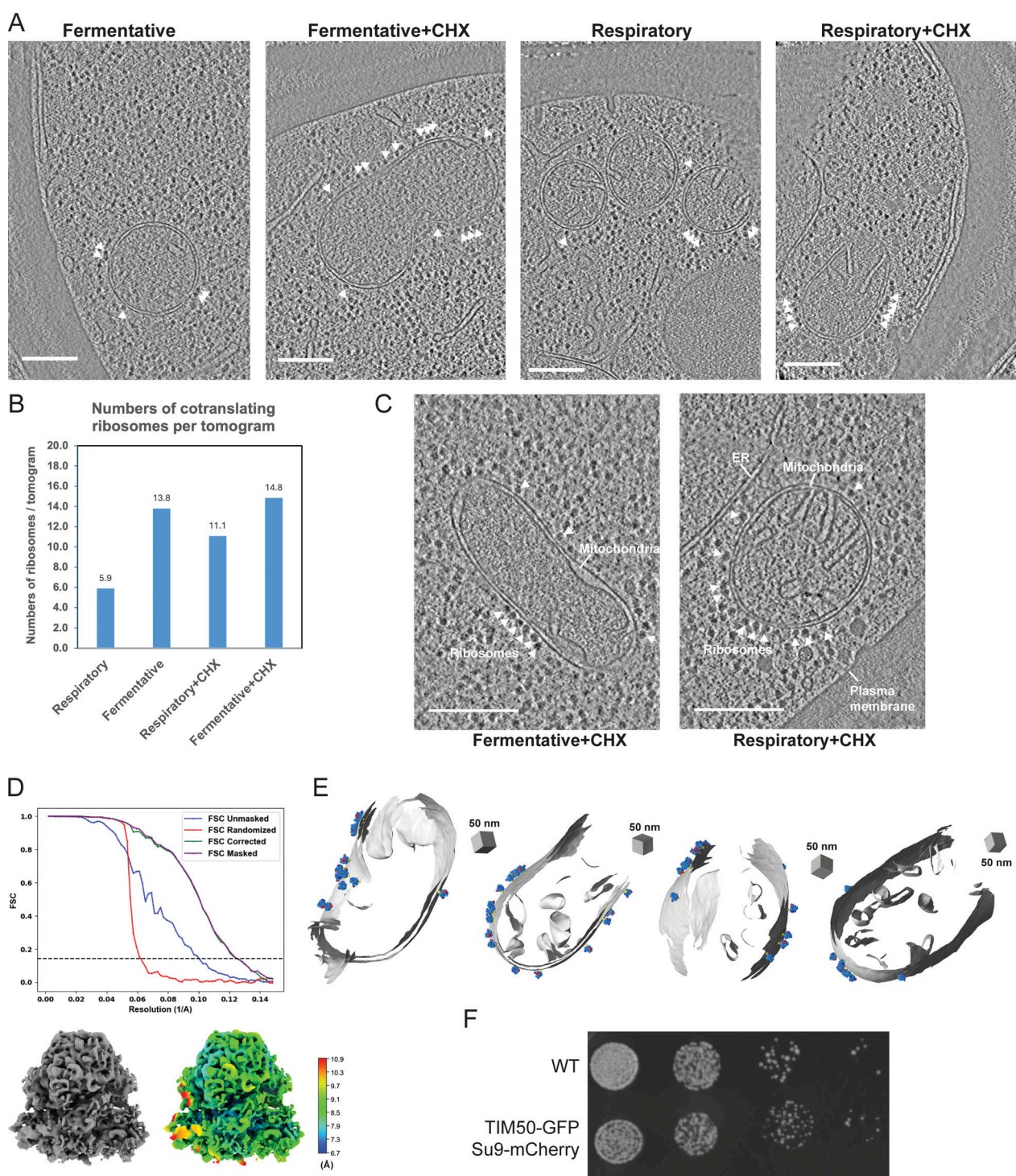

Figure S1.  **Representative tomograms of cryo-FIB milled S. cerevisiae cell lamellae showed visible mitochondria-associated cytoplasmic ribosomes.** **(A)** Representative X-Y slices of reconstructed tomograms collected at pixel size 2.638 Å from cryo-FIB milled S. cerevisiae yeast cells grown in different growth conditions (i.e., fermentative and respiratory) and treatment conditions (i.e., vehicle and CHX). Cytoplasmic ribosomes in close proximity to the OMM are highlighted by white arrowheads. Scale bars = 250 nm. **(B)** Quantification of the number of ribosomes positioned with the exit tunnel facing the OMM in CHX-treated or vehicle-treated cells grown in respiratory versus fermentative conditions. **(C)** Representative X-Y slices of reconstructed tomograms collected at pixel size 1.6626 Å from cryo-FIB milled S. cerevisiae grown in fermentative and respiratory conditions and treated with CHX (100 µg/ml) displaying subcellular features such as mitochondria, ribosomes, the ER, and the plasma membrane. Cytoplasmic ribosomes in close proximity to the OMM are highlighted by white arrowheads. Scale bars = 250 nm. **(D)** FSC plot (top) of the 80S cytoplasmic ribosome reconstruction is shown with resolution reported at 0.143 FSC and the reconstructed subtomogram average (bottom left). The 80S cytoplasmic ribosome was resolved to 8 Å from 35,784 ribosome particles with the color map (bottom right) showing the local resolution. **(E)** A subset of representative models of ribosomes positioned with their exit tunnels optimally positioned for protein import at the OMM surface. **(F)** Serial 10-fold dilutions of fluorescently-labeled mitochondrial strains show similar growth to the parent WT (BY4741) cells on a non-fermentable carbon source YPGE (Yeast extract, Peptone, 3% Glycerol, 2% Ethanol).

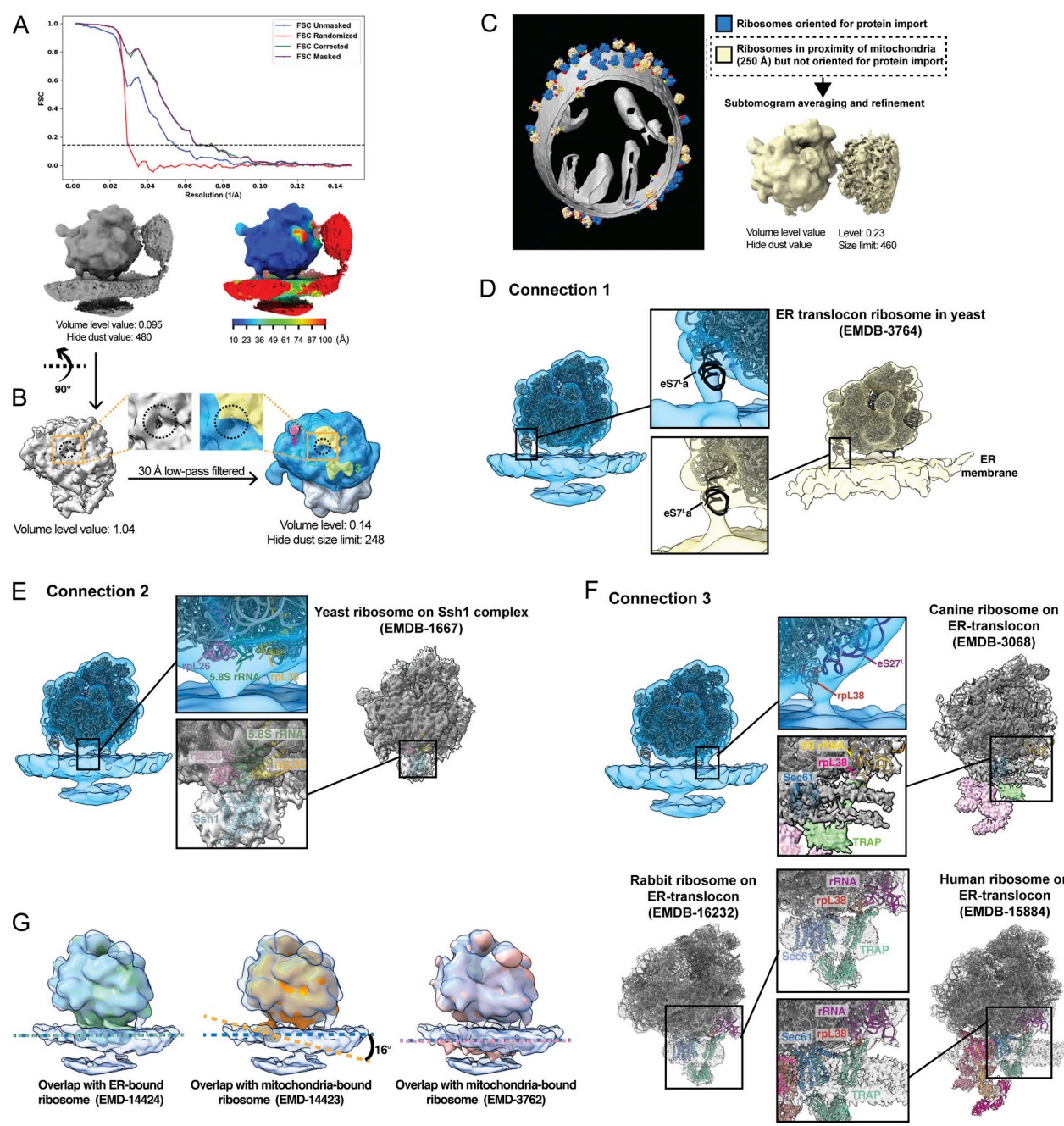

Figure S2. **3D subtogram average of a cytoplasmic ribosome positioned for protein import on the OMM shows multiple contact points. (A)** FSC plot of the OMM-associated 80S cytoplasmic ribosome reconstruction is shown with resolution reported at 0.143 FSC and the reconstructed subtogram average (bottom left). The 80S cytoplasmic ribosome was resolved to 19 Å from 1,076 ribosome particles with the color map shows the local resolution (bottom right). **(B)** The peptide exit tunnel is visible in the subtogram average (gray density) at higher isosurface volume thresholds, as indicated by the dashed black circle. This was used to mark its position relative to the connecting densities visible in the subtogram average at lower isosurface volume threshold values (colored density). **(C)** Representative models of ribosomes positioned within 250 Å of the OMM surface with their exit tunnels facing away from the OMM. 3D refinement of these particles results in a 3D reconstruction of a ribosome that does not contain any distinguishable connecting densities between the 80S ribosome and the OMM, suggesting that these connections are specific to 80S ribosomes optimally positioned for protein import. **(D)** Density corresponding to the connection labeled #1 in the subtogram average of mitochondrial-associated ribosomes correlates well with the density corresponding to the expansion segment of eS7La of the 25S rRNA in the large 60S subunit present in the ER-associated ribosome maps from *S. cerevisiae* (EMDB-3764). **(E)** The density corresponding to the connection labeled #2 correlates well with the density corresponding to the region where the ribosome interacts with the import channel, Ssh1, in the ER-associated ribosome maps from *S. cerevisiae* (EMD-1667). **(F)** The density corresponding to the connection labeled #3 correlates well with the density corresponding to the rRNA expansion segment, rpL38, and the TRAP complex in the ER-associated ribosome maps from human (EMD-15884), rabbit (EMD-16232), and canine (EMD-3068). **(G)** Overlap between the OMM-associated 80S cytoplasmic ribosome structure from this work (transparent blue) with ER-bound ribosome (EMD-14424; green) and mitochondria-bound ribosome (EMD-14423, orange) from *S. pombe* and mitochondria-bound ribosome from purified *S. cerevisiae* mitochondria (EMD-3762, pink). Dashed colored lines show the relative orientations of the membranes for the different structures.

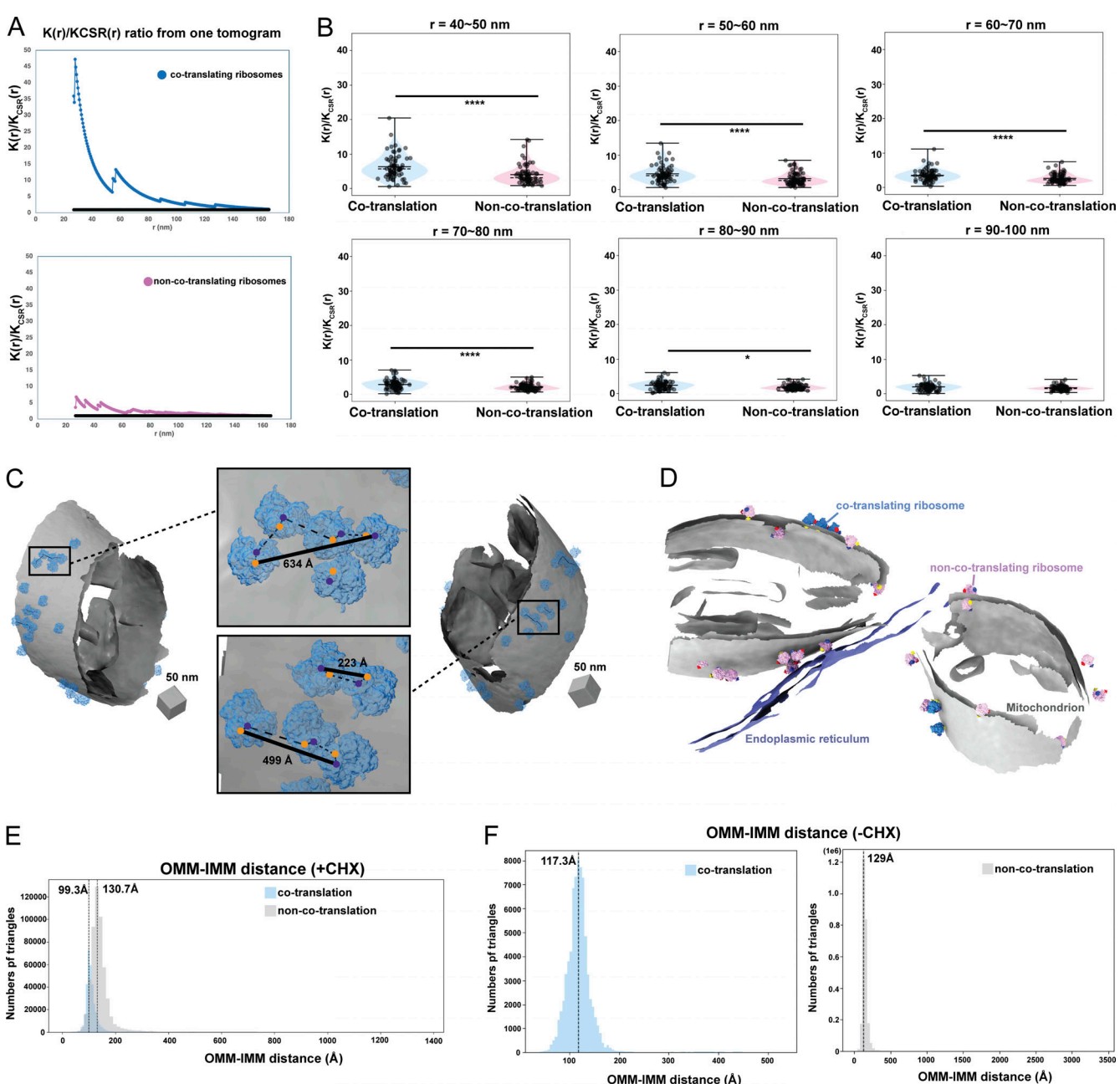

Figure S3. **Cytoplasmic ribosomes primed for protein import cluster on the mitochondrial membrane. (A)** Representative plots of the K(r)/K$_{CSR}$(r) ratio for mitochondrial-associated cytoplasmic ribosomes oriented for protein import within a range of radius (r) values of 27–166 nm. The black line equals to 1. **(B)** Quantification of the maximum value of K(r)/K$_{CSR}$(r) for each tomogram at the indicated radius intervals for each ribosome class. Quantification from import-oriented ribosomes *n* = 87 and non-import-oriented ribosomes *n* = 89 tomograms are shown. P values from Mann–Whitney U test are indicated. *P < 0.05; ****P < 0.001. **(C)** Representative membrane surface reconstructions of mitochondria (gray) with ribosomes oriented for import relative to the OMM (blue). Insets show zoomed-in boxed regions of the ribosome models with circle overlays demarking the location of the 3′ mRNA entry (blue), the 5′ mRNA exit sites (orange), the possible pathways of interconnecting mRNA (dashed black line), and the calculated end-to-end distance from 5′ to 3′ of each interconnected mRNA (solid black line). **(D)** Membrane surface reconstruction of mitochondria (gray) and ER (purple) membranes with corresponding models for co-translating (blue) and non-co-translating ribosomes (pink). **(E)** Combined histogram of IMM–OMM distances of co-translation-associated and non-co-translation-associated patches in *S. cerevisiae* treated with CHX. Dashed vertical lines correspond to peak histogram values of pooled data. **(F)** Histograms of IMM–OMM distances of co-translation-associated and non-co-translation-associated patches in *S. cerevisiae* treated with vehicle (e.g., no CHX). Dashed vertical lines correspond to peak histogram values of pooled data.

Video 1.   **3D subtomogram average of a cytoplasmic ribosome optimally positioned for protein import on the OMM shows multiple contact points.** A subtomogram average of the import-oriented cytoplasmic ribosome forms three distinct connections (connection #1, 2, 3) to OMM under CHX treatment. A model of the *S. cerevisiae* 80S ribosome structure (PDB 4V6I) is rigid-body fit to highlight the ribosomal components involved in each connection. Connection #1 overlaps with eS7La (light blue). Connection #2 overlaps with rpL26 (pink), rpL35 (yellow), and 5.8S rRNA (green) near the ribosomal peptide exit tunnel (blue sphere). Connection #3 overlaps with rpL38 (light yellow) and eS27L (red).

