## [Peer Review File · The Journal of Cell Biology]

Cytoplasmic ribosomes on mitochondria alter the local membrane environment for protein import

Ya-Ting Chang, Benjamin Barad, Juliette Hamid, Hamidreza Rahmani, Brian Zid, and Danielle Grotjahn

Corresponding Author(s): Danielle Grotjahn, Scripps Research Institute

Review Timeline:

Submission Date:	2024-07-17
Editorial Decision:	2024-10-08
Revision Received:	2024-12-04
Editorial Decision:	2024-12-17
Revision Received:	2025-01-17

Monitoring Editor: Jodi Nunnari

Scientific Editor: Dan Simon

Transaction Report:

DOI: <https://doi.org/10.1083/jcb.202407110>

October 8, 2024

Re: JCB manuscript #202407110

Dr. Danielle Ann Grotjahn
Scripps Research Institute
Department of Integrative Structural and Computational Biology
10550 North Torrey Pines Rd
Hazen-113
La Jolla, California 92037

Dear Dr. Grotjahn,

Thank you for submitting your manuscript entitled "Cytoplasmic ribosomes on mitochondria alter the local membrane environment for protein import." Thank you for your patience with the peer review process. The manuscript was assessed by expert reviewers, whose comments are appended to this letter. We invite you to submit a revision if you can address the reviewers' key concerns, as outlined here.

You will see that the reviewers are enthusiastic about your study and state that it is well done and provides interesting new insights into the process of co-translational import into mitochondria. They do, however, raise important points that need to be addressed. Reviewers #1 & 3 ask to explain the choice of the strain that overexpresses Tim50 as well as to confirm that it does not have import defects and that ribosome association with mitochondria is not induced by Tim50. Reviewer #1 also asks for evidence that mitochondria associated ribosomes are productive rather than stalled. While Reviewer #2 appreciated the proposed model in Figure S4D, Reviewers #1 & 3 felt that the fitting of TOM40 and NAC complexes into the membrane density was too speculative at this stage. We agree that in the absence of additional data confirming the proposed interactions of these complexes with ribosomes, this model should be removed from the manuscript. The other comments ask for method clarifications, improved explanations of data and conclusions, and a few discussion points. These all seem fairly straightforward and should be addressed by text and figure revisions.

GENERAL GUIDELINES:

Text limits: Character count for a Report is < 20,000, not including spaces. Count includes title page, abstract, introduction, the joint Results & Discussion, and acknowledgments. Count does not include materials and methods, figure legends, references, tables, or supplemental legends.

Figures: Reports may have up to 5 main text figures. To avoid delays in production, figures must be prepared according to the policies outlined in our Instructions to Authors, under Data Presentation, <https://jcb.rupress.org/site/misc/ifora.xhtml>. All figures in accepted manuscripts will be screened prior to publication.

*****IMPORTANT:** It is JCB policy that if requested, original data images must be made available. Failure to provide original images upon request will result in unavoidable delays in publication. Please ensure that you have access to all original microscopy and blot data images before submitting your revision. *******

Supplemental information: There are strict limits on the allowable amount of supplemental data. Reports may have up to 3 supplemental figures. Up to 10 supplemental videos or flash animations are allowed. A summary of all supplemental material should appear at the end of the Materials and methods section.

Please note that JCB now requires authors to submit Source Data used to generate figures containing gels and Western blots with all revised manuscripts. This Source Data consists of fully uncropped and unprocessed images for each gel/blot displayed in the main and supplemental figures. If your revised manuscript will include cropped gel and/or blot images, please be sure to provide one Source Data file for each figure that contains gels and/or blots along with your revised manuscript files. File names for Source Data figures should be alphanumeric without any spaces or special characters (i.e., SourceDataF#, where F# refers to the associated main figure number or SourceDataFS# for those associated with Supplementary figures). The lanes of the gels/blots should be labeled as they are in the associated figure, the place where cropping was applied should be marked (with a box), and molecular weight/size standards should be labeled wherever possible. Source Data files will be made available to reviewers during evaluation of revised manuscripts and, if your paper is eventually published in JCB, the files will be directly linked to specific figures in the published article.

Source Data Figures should be provided as individual PDF files (one file per figure). Authors should endeavor to retain a

minimum resolution of 300 dpi or pixels per inch. Please review our instructions for export from Photoshop, Illustrator, and PowerPoint here: <https://rupress.org/jcb/pages/submission-guidelines#revised>

The typical timeframe for revisions is three to four months. If you anticipate any difficulties in meeting this aforementioned revision time limit, please contact us and we can work with you to find an appropriate time frame for resubmission. Please note that papers are generally considered through only one revision cycle, so any revised manuscript will likely be either accepted or rejected.

Thank you for this interesting contribution to Journal of Cell Biology. You can contact us at the journal office with any questions at cellbio@rockefeller.edu.

Sincerely,

Jodi Nunnari, PhD
Editor-in-Chief
Journal of Cell Biology

Dan Simon, PhD
Scientific Editor
Journal of Cell Biology

Reviewer #1 (Comments to the Authors (Required)):

Most mitochondrial proteins are synthesized by cytosolic ribosomes. Mitochondrial proteins can be imported post- or co-translationally depending on the timing of protein synthesis and protein import. In vitro, most mitochondrial precursor proteins can be imported independently of their synthesis, but in vivo, synthesis and translocation might be coupled, at least for certain mitochondrial proteins.

In this study, the authors used cryo electron tomography (cryo ET) to study the orientation and distribution of mitochondria-associated cytosolic ribosomes in yeast cells. They observed that a small number of ribosomes was associated with the mitochondrial surface and positioned with the exit tunnel towards the membrane. The orientation of these ribosomes suggest that polysomes associate with the mitochondrial surface. The authors claim that ribosomes 'touch' the outer membrane with three contact points constituted by eS7a, 5.8 rRNA and rpL38. Areas where bound cytosolic ribosomes show a reduced distance between the outer and inner membrane indicating contact sites or zones between both mitochondrial membranes. Finally, cytosolic ribosomes were not found to be enriched around ER mitochondria contact sites as only few of such contacts were found in the analyzed samples.

The study is technically impressive. In particular, the identification of the membranes in the sections is very compelling and only possible due to recent developments in image analysis. The aspect of co-translational import addresses a 'hot topic' in the field as it is unclear to what extent mitochondrial proteins are imported during their synthesis. Still, the advance in understanding from this study is somewhat limited. The authors correctly cite a study published in JCB 50 years ago (!) which showed the presence of cytosolic ribosomes on the mitochondrial surface, showed that cycloheximide treatment increased this population and showed that ribosomes were binding at zones of close contact of outer and inner membranes. Thus, the conclusions of this old study are almost the same as those of this study now. Moreover, a more recent study by Vicki Gold and Werner Kuhlbrandt had shown cytosolic ribosomes on isolated yeast mitochondria. Nevertheless, this study now is the first to use cryo ET thus showing mitochondria in situ and the resolution is now much much better. I am convinced that cryo-ET will be very powerful to solve the issue of co-translational import into mitochondria and this study is timely and inspiring. Therefore, I am convinced that this is an interesting study of high technical quality, and the Journal of Cell Biology seems to me well suited for such an article. A couple of points need to be addressed.

1. The authors use a strain for their analysis which overexpresses Tim50-GFP. This is a problem. Tim50 is one of the few proteins which cannot be imported post-translationally into mitochondria (Yamamoto et al. 2002 Cell 111, 519f). One of the authors (Brian Zid) previously published evidence that the TIM50 mRNA is bound to mitochondria and suggested that a translational stalling mechanism might recruit ribosome-nascent chain complexes to mitochondria. Thus, the expression of Tim50, in particular when fused to GFP, might artificially increase the number of cytosolic ribosomes in cells. The authors therefore have to show that the ribosome-association in their study does not depend on the Tim50-GFP expression construct. The same applies to the observed effects on membrane spacing.

2. The authors claim that ribosome-associated protein import alters the local architecture of the outer and inner membrane. Also the title implies that ribosomes alter the membrane environment. However, there is little evidence that ribosomes are involved

here. It is well established in the field, that the TOM and TIM23 translocases dynamically associate during translocation of matrix proteins (see for example Chacinska et al. 2003 EMBO 22, 5370f). Thus, the close contact of the membrane in ribosome-associated zones might simply be the consequence of translocating nascent chains that are arrested by cycloheximide.

3. The authors speculate about the relevance of NAC and even show a hypothetical structure including NAC in the supplement. The study would be more compelling if structures from a NAC-deficient mutant were shown.

4. The authors speculate about co-translating ribosomes. At least in the structures shown in Fig. S6, the ribosomes seem to be extremely close to each other and resemble more the situation of collided ribosome. Is there any evidence that these ribosomes are productive rather than stalled ribosomes which await to be rescued by RQC?

5. A recent study cryo-ET study on human cells reported major differences in the membrane orientation of mitochondria- and ER-bound ribosomes (de Teresa-Trueba et al. 2023 Nature Methods 20, 284f, Extended Data Fig. 9). The potential differences to this study should be discussed.

6. The authors have to clearly point out which of their structures are based on cycloheximide-stalled conditions. Currently, this is not easy to extract from the descriptions of the experiments. As such conditions are highly artificial, the interpretation has to be done with caution. 'Productive' ribosomes might be very different to those that are artificially stalled by cycloheximide. It is important to show physiological conditions for comparison, even if they are at lower resolution.

Reviewer #2 (Comments to the Authors (Required)):

Chang and colleagues performed a tomographic study to characterize translating cytosolic ribosomes on the mitochondrial surface. Treatment of the cells with cycloheximide increases the number of cytosolic ribosomes on mitochondria. A portion of them orients in such a manner that allows co-translational import. The authors found that these ribosomes form three contact sites to mitochondria. One of them could comprise the TOM complex, while another one could provide space for NAC to bind. Ribosomes cluster on mitochondria and might be arranged in polysomes. The quality of the presented data is high and the findings are well described. The structural insights into operating ribosomes at mitochondria are impressive.

Most of these findings were already obtained with isolated mitochondria (Gold et al., 2017), supporting the obtained data of this study. The authors nicely compared their findings to the previous study. The progress in the present study is that the cryo-electron tomography provides the first insights about translating ribosomes in cells. Furthermore, this study showed that translating ribosome decrease the distance between outer and inner membrane, which was not shown before. Therefore, the work is interesting for a broad readership of Journal of Cell Biology.

Minor comment:

I suggest to move Supplementary Figure 4D into the main figure. It shows an interesting model how the ribosome may dock on the TOM complex. Is it possible and reasonable to model a precursor protein in this model?

Reviewer #3 (Comments to the Authors (Required)):

The authors take a membrane-guided approach to identify cytoplasmic ribosomes oriented for protein import on the mitochondrial surface in *Saccharomyces cerevisiae* using cryo-electron tomography (cryo-ET). The authors show that ribosomes cluster, make multiple contacts with, and induce local changes to the mitochondrial membrane ultrastructure at import sites. Overall, this is timely study that applies cryo-ET and new analytic methods developed by the authors to address an interesting question in mitochondrial protein import. The findings contradict the prevailing view that nuclear-encoded mitochondrial proteins are always imported post-translationally.

One of the more interesting observations is that at the putative co-translational import sites, the authors found ribosome clustering that coincide with striction of the OMM-IMM distance. The authors note that this would facilitate hand-off of protein import substrates from TOM to TIM complexes, but this should be discussed a little further. What are the cytosolic ribosomes actually doing to alter membrane geometry? Could it be that the nascent polypeptide itself is responsible for tethering the OMM-IMM and narrowing the IM space?

The TOM complex docking in Fig. S4D is overlay speculative and should be removed unless stronger evidence for the docking mode can be obtained.

It was unclear why a GFP-TIM50 yeast strain was used. CLEM seems to have been used simply to identify yeast cells, nothing more. Has this strain been validated as non-perturbing for import?

How confident are the authors that no ER-bound ribosomes are being picked and incorporated into the sub-tomo average? These would look nearly identical based on the spherical mask chosen.

The authors have identified many ribosomes on the surface of the OMM that are not primed for import. What fraction of

ribosomes are in the right orientation for import? What is the function of the ribosomes on the membrane in the wrong orientation? Could they just be nearby cytoplasmic ribosomes that aren't actually interacting with the OMM at all?

RESPONSE TO EDITOR/REVIEWER COMMENTS

Editorial Comments:

Thank you for submitting your manuscript entitled "Cytoplasmic ribosomes on mitochondria alter the local membrane environment for protein import." Thank you for your patience with the peer review process. The manuscript was assessed by expert reviewers, whose comments are appended to this letter. We invite you to submit a revision if you can address the reviewers' key concerns, as outlined here.

You will see that the reviewers are enthusiastic about your study and state that it is well done and provides interesting new insights into the process of co-translational import into mitochondria. They do, however, raise important points that need to be addressed. Reviewers #1 & 3 ask to explain the choice of the strain that overexpresses Tim50 as well as to confirm that it does not have import defects and that ribosome association with mitochondria is not induced by Tim50. Reviewer #1 also asks for evidence that mitochondria associated ribosomes are productive rather than stalled. While Reviewer #2 appreciated the proposed model in Figure S4D, Reviewers #1 & 3 felt that the fitting of TOM40 and NAC complexes into the membrane density was too speculative at this stage. We agree that in the absence of additional data confirming the proposed interactions of these complexes with ribosomes, this model should be removed from the manuscript. The other comments ask for method clarifications, improved explanations of data and conclusions, and a few discussion points. These all seem fairly straightforward and should be addressed by text and figure revisions.

We thank the editor for the positive response to our manuscript and the reviewer's careful and thorough read of our initial manuscript. We address all the reviewer's valuable comments in this revised manuscript, as outlined below. We feel that the manuscript has been significantly improved by the review process at *JCB* and we thank you and the reviewer's for their time and effort in helping improve our submission.

Reviewer #1 (Comments to the Authors (Required)):

Most mitochondrial proteins are synthesized by cytosolic ribosomes. Mitochondrial proteins can be imported post- or co-translationally depending on the timing of protein synthesis and protein import. In vitro, most mitochondrial precursor proteins can be imported independently of their synthesis, but in vivo, synthesis and translocation might be coupled, at least for certain mitochondrial proteins.

In this study, the authors used cryo electron tomography (cryo ET) to study the orientation and distribution of mitochondria-associated cytosolic ribosomes in yeast cells. They observed that a small number of ribosomes was associated with the mitochondrial surface and positioned with the exit tunnel towards the membrane. The orientation of these ribosomes suggest that polysomes associate with the mitochondrial surface. The authors claim that ribosomes 'touch' the outer membrane with three contact points constituted by eS7a, 5.8 rRNA and rpL38. Areas where bound cytosolic ribosomes show a reduced distance between the outer and inner membrane indicating contact sites or zones between both mitochondrial membranes. Finally, cytosolic ribosomes were not found to be enriched around ER mitochondria contact sites as only few of such contacts were found in the analyzed samples.

The study is technically impressive. In particular, the identification of the membranes in the sections is very compelling and only possible due to recent developments in image analysis. The aspect of co-translational import addresses a 'hot topic' in the field as it is unclear to what extent mitochondrial proteins are imported during their synthesis. Still, the advance in understanding from this study is somewhat limited. The authors correctly cite a study published in *JCB* 50 years ago (!) which showed the presence of cytosolic ribosomes on the mitochondrial surface, showed that cycloheximide treatment increased this population and showed that ribosomes were binding at zones of close contact of outer and inner membranes. Thus, the conclusions of this old study are

almost the same as those of this study now. Moreover, a more recent study by Vicki Gold and Werner Kuhlbrandt had shown cytosolic ribosomes on isolated yeast mitochondria. Nevertheless, this study now is the first to use cryo ET thus showing mitochondria in situ and the resolution is now much much better. I am convinced that cryo-ET will be very powerful to solve the issue of co-translational import into mitochondria and this study is timely and inspiring. Therefore, I am convinced that this is an interesting study of high technical quality, and the Journal of Cell Biology seems to me well suited for such an article. A couple of points need to be addressed.

1. The authors use a strain for their analysis which overexpresses Tim50-GFP. This is a problem. Tim50 is one of the few proteins which cannot be imported post-translationally into mitochondria (Yamamoto et al. 2002 Cell 111, 519f). One of the authors (Brian Zid) previously published evidence that the TIM50 mRNA is bound to mitochondria and suggested that a translational stalling mechanism might recruit ribosome-nascent chain complexes to mitochondria. Thus, the expression of Tim50, in particular when fused to GFP, might artificially increase the number of cytosolic ribosomes in cells. The authors therefore have to show that the ribosome-association in their study does not depend on the Tim50-GFP expression construct. The same applies to the observed effects on membrane spacing.

We would like to point out that there is no overexpression of Tim50-GFP in this strain. This strain has GFP integrated at the endogenous locus of Tim50 under the control of the endogenous Tim50 promoter. This strain shows similar growth as a WT BY4741 strain in fully respiratory metabolism YPGlycerolEthanol (now included as **Supplementary Figure 1F**), implying no substantial mitochondrial import defects. We also confirmed through cryo-fluorescence microscopy that mitochondria morphology remains tubular based on the Su9-mCherry signal (Figures below), consistent with previous reports (Figure 3, PMID: 11978537). Based on our tomographic data and extrapolating for the entire volume of a yeast cell, we estimate there will be >100 co-importing ribosomes in an average cell under cycloheximide treatment. Since TIM50 has less than 5 mRNAs/cell on average, it is extremely unlikely that TIM50-GFP expression could lead to all the co-translating ribosome events we observe in our tomographic data. Collectively, these data suggest that the Tim50-GFP strain is unlikely to induce defects respiratory growth, mitochondrial morphology, and co-translational import events we observe in the present study. We have clarified this point by adding the respiratory growth data to **Supplementary Figure 1F** and further described this cell line in **lines 283-287 and 303-305** in the 'Materials and Methods' section.

*“We verified that fluorescence labels did not impact mitochondrial health by performing a growth assay of cells from wild-type (BY4741) and fluorescently-labeled strains (Su9-mCherry, TIM50-GFP). Cells were grown to the logarithmic phase in YPG media, and serial 10-fold dilutions of the same starting OD were then spotted (10 μL) onto YPG respiratory plates and incubated at 30°C (**Supplementary Figure 1F**).”*

“We also assessed mitochondrial network morphology using the fluorescence signal from Su9-mCherry to evaluate mitochondrial integrity.”

2. The authors claim that ribosome-associated protein import alters the local architecture of the outer and inner membrane. Also the title implies that ribosomes alter the membrane environment. However, there is little

evidence that ribosomes are involved here. It is well established in the field, that the TOM and TIM23 translocases dynamically associate during translocation of matrix proteins (see for example Chacinska et al. 2003 EMBO 22, 5370f). Thus, the close contact of the membrane in ribosome-associated zones might simply be the consequence of translocating nascent chains that are arrested by cycloheximide.

We agree with the reviewer that the close contact observed in the ribosome-associated zones might result from the translocating nascent chains associated with the ribosome. However, we would like to emphasize that this effect appears independent of cycloheximide arrest, as we observe a similarly reduced OMM-IMM distance in untreated cells (absence of cycloheximide) (Fig 3C). Additionally, we suggest that the close contact may not solely result from translocating nascent chains. A previous study (Table 3, PMID: 24942077), which identified the location of quantum-dot labeled import supercomplexes, such as TOM-TIM23, and measured the OMM-IMM distance in importing mitochondria, showed no significant difference in OMM-IMM spacing between importing and non-importing mitochondria. This comparison suggests that the close contact we observed may instead be associated with ribosome-mediated protein import. We have added additional text to clarify this point in **lines 237-240** of the revised manuscript:

“Despite notably fewer ribosomes optimally positioned for protein import in the vehicle-treated (i.e., non-CHX-treated) condition, we observed a similar decrease in the intermembrane distance in “co-translation-associated” patches relative to “non-co-translation-associated” patches (Figure 4B), suggesting the close contact in “co-translation-associated” patches is independent of translational arrest”

3. The authors speculate about the relevance of NAC and even show a hypothetical structure including NAC in the supplement. The study would be more compelling if structures from a NAC-deficient mutant were shown.

We agree that this is exciting, and our cryo-ET system is now set up to perform these mechanistic and structure-mapping studies in the future. However, as the primary aim of this manuscript is to present the first high-resolution views of mitochondrial co-translation with cryo-ET, we believe this topic falls beyond its current scope.

4. The authors speculate about co-translating ribosomes. At least in the structures shown in Fig. S6, the ribosomes seem to be extremely close to each other and resemble more the situation of collided ribosome. Is there any evidence that these ribosomes are productive rather than stalled ribosomes which await to be rescued by RQC?

While our analysis indicates that ribosome clusters may resemble the orientation of polysomes, further experiments are needed to confirm if these clusters are on the same mRNA. At this stage, we cannot determine whether these are collided ribosomes awaiting RQC rescue without evidence of mRNA connection. We agree there could be some collided ribosomes, and we have even previously published that ribosomal slowdown is beneficial for cotranslational targeting of the TIM50 mRNA to mitochondria (PMID: 32762840). We would also like to note that the CHX treatment is at a level to freeze ribosome movement, used in ribosome profiling experiments (PMID: 19213877), and not the intermediate levels of translation elongation inhibitors used to artificially induce ribosome collisions (PMID: 32610081). Additionally, seeing the local alteration of membrane distance under import-oriented ribosomes further indicates most of them are translocating proteins. Therefore, they are more likely productive rather than stalled ribosomes. Although examining ribosomes undergoing RQC at the mitochondrial surface is an intriguing avenue for future research, we feel it falls outside the scope of the present manuscript.

5. A recent study cryo-ET study on human cells reported major differences in the membrane orientation of mitochondria- and ER-bound ribosomes (de Teresa-Trueba et al. 2023 Nature Methods 20, 284f, Extended Data Fig. 9). The potential differences to this study should be discussed.

We thank the reviewer for bringing this to our attention. In de Teresa-Trueba et al. 2023, the authors note there is a 33 to 35-degree difference in the orientation of the membrane between the ER-bound and the mito-bound structures from their data. In comparing our structure to the structures of both the mito-bound and ER-bound ribosome in previous work, we observe that our structure more closely resembles the ER-bound ribosome structure (see Figure A below). When we align the ribosome portions of both our structure and the mito-bound ribosome structure from de Teresa-Trueba et al, we observe a slight 16-degree shift in the position of the membrane (see Figure B below). We propose three possible explanations for this discrepancy:

1. Differences in the orientation of selected ribosome populations used for subtomogram averaging: In de Teresa-Trueba et al. 2023, the authors identified mitochondria-bound ribosomes solely based on their proximity to the mitochondria without considering the ribosome's orientation. As a result, the population of mitochondria-bound ribosomes in that study likely included a mixture of co-translating and non-co-translating ribosomes. In our study, we used our morphometrics approach to specifically identify the population of ribosomes likely performing co-translational import on the basis that they are both in close proximity to (0-250 Å) and oriented with their peptide exit tunnel facing the outer mitochondrial membrane. While we allowed certain degrees of freedom for the ribosome orientation, most ribosomes that were in close proximity to but not oriented for import on the OMM were excluded. This mixing of ribosome populations could have influenced the observed differences in membrane orientation between mitochondria-bound and ER-bound ribosomes.
2. Differences in box size of the ribosome 'particles' used for subtomogram averaging: When examining mito-bound structure from previous work (EMD-14423) we noticed that the particle 'box' size they used for extraction and averaging is slightly smaller than the box size used in our structure. This is also evident in the fact that very little membrane is resolved in the mito-bound ribosome structure from previous work, making it difficult to interpret the exact position of the membrane. In our structure, we used a slightly larger box size, which enabled us to capture more of the outer mitochondrial membrane and even some density spanning into what we interpret is likely the inner membrane space.
3. Species-specific or CHX-dependent differences: Our structure and Gold et al. 2017 structures of mitochondria-bound ribosomes both come from CHX-treated *Saccharomyces cerevisiae* cells, whereas de Teresa Trueba et al structure is from *Schizosaccharomyces pombe*. In the Gold et al 2017 paper, both the ER-associated and the mitochondria-associated ribosomes show the same membrane orientation. Therefore, the differences observed in de Teresa Trueba et al ER- and mito-associated ribosomes may be specific to *Schizosaccharomyces pombe*, whereas *Saccharomyces cerevisiae* retain the same membrane orientation. Furthermore, in agreement with our structure, the Gold et al 2017 paper using cryo-ET on purified mitochondria from CHX-treated *S. cerevisiae* showed the mito-bound ribosome positioned similar to ours with respect to the outer mitochondrial membrane, albeit no connections between the ribosome and membrane were resolved in this work (see Figure C below). While we need to interpret our structure in the context of translation elongation (i.e., CHX-treatment), we believe our structure better captures what is likely the most stable 'docking' position of cytoplasmic ribosomes on the outer mitochondria membrane.

4.

To address this point further, we included an additional figure panel (**Supplementary Figure 2G**) and manuscript text (**lines 143-154**) to discuss these differences:

*“Previous studies report a 33-35° difference in the membrane orientation between ER-associated (EMD-14424) and mitochondria-associated ribosome (EMD-14423) structures in *S. pombe* cells. In contrast, we observe no difference in the alignment of our mitochondria-associated ribosome structure to the ER-associated structure and only a slight 16° difference in the orientation of the mitochondria-associated structure from *S. pombe* (**Supplementary Figure 2G**). These variations could be due to differences in the orientation of selected ribosome populations used for subtomogram averaging, as our method considers both ribosome distance and orientation relative to the OMM instead of distance alone. Additionally, these differences could reflect the accuracy in capturing membrane positioning due to discrepancies in the size of the ‘particle box size’ used for computational extraction and averaging, resulting in additional membrane density visible in our subtomogram average relative to previous reports (de Teresa-Trueba, Goetz et al. 2023). Finally, these structural differences may be CHX-dependent or species-specific, as no orientation differences were observed between ER- and mitochondria-associated ribosomes in CHX-treated *S. cerevisiae* cells (Gold, Chroscicki et al. 2017).”*

6. The authors have to clearly point out which of their structures are based on cycloheximide-stalled conditions. Currently, this is not easy to extract from the descriptions of the experiments. As such conditions are highly artificial, the interpretation has to be done with caution. 'Productive' ribosomes might be very different to those that are artificially stalled by cycloheximide. It is important to show physiological conditions for comparison, even if they are at lower resolution.

We appreciate the reviewer's point and fully agree that it is crucial to critically assess the potential effects of CHX when interpreting the biological findings resulting from this treatment condition. We want to clarify that all structural data except for Figure 4B is from CHX-treated cells. Similar to previous work (PMID: 28827470), we noted that cells treated with CHX give substantially more mitochondrial-associated ribosomes (n=1,076) relative to non-CHX-treated cells (n=185), which likely helped to increase the resolution and clarity of our final reconstruction. We agree that it is worth exploring physiological conditions; however, this will likely require substantially increasing our datasets (~10x) and, therefore, beyond the scope of this present manuscript. Nonetheless, per the reviewer's suggestion, we averaged the 185 ribosomes oriented for import in untreated (i.e., non-CHX treated) datasets. We observed that the ribosome structure in untreated datasets resembles a similar orientation to the membrane as the ribosome structure in CHX-treated datasets (Figure below). While we

only observe a single connection in our preliminary structure of non-CHX treated datasets (closest to connection #3), this difference needs to be interpreted with extreme caution. Given that the features visible in the subtomogram averaged structure depend on particle numbers and interaction stability, we suggest that the fewer contact sites could reflect a lower number of import-oriented ribosomes captured—approximately ten times fewer than in CHX-treated datasets. Since it is difficult to resolve whether these differences are due to technical (i.e., lack of particles) or biological (i.e., CHX-induced differences), we prefer to position this current manuscript in the context of CHX treatment. However, we agree with the reviewer that future work is needed to determine whether the connections we observe are influenced by translation elongation inhibition or replicate what would be considered a more ‘productive’ state. We have further clarified in several places in our text that the structures in this paper are within the context of CHX treatment (**lines 99, 111, 153, 156, 163, 265**) and added the following statement in **lines 155-157**:

“In summary, our structure shows density for three contacts between the cytoplasmic ribosome and the OMM in cells treated with CHX, which share structural similarities with ER-ribosome interactions during ER co-translational import.”

Reviewer #2 (Comments to the Authors (Required)):

Chang and colleagues performed a tomographic study to characterize translating cytosolic ribosomes on the mitochondrial surface. Treatment of the cells with cycloheximide increases the number of cytosolic ribosomes on mitochondria. A portion of them orients in such a manner that allows co-translational import. The authors found that these ribosomes form three contact sites to mitochondria. One of them could comprise the TOM complex, while another one could provide space for NAC to bind. Ribosomes cluster on mitochondria and might be arranged in polysomes. The quality of the presented data is high and the findings are well described. The structural insights into operating ribosomes at mitochondria are impressive.

Most of these findings were already obtained with isolated mitochondria (Gold et al., 2017), supporting the obtained data of this study. The authors nicely compared their findings to the previous study. The progress in the present study is that the cryo-electron tomography provides the first insights about translating ribosomes in cells. Furthermore, this study showed that translating ribosome decrease the distance between outer and inner membrane, which was not shown before. Therefore, the work is interesting for a broad readership of Journal of Cell Biology.

Minor comment:

I suggest to move Supplementary Figure 4D into the main figure. It shows an interesting model how the ribosome may dock on the TOM complex. Is it possible and reasonable to model a precursor protein in this model?

We agree that the model presented in Supplementary Figure 4D is interesting. However, it appears too speculative for other reviewers, so we will remove it based on their feedback. In response to the second question, we believe it is not feasible to model the precursor protein due to the limited resolution of our structure.

Reviewer #3 (Comments to the Authors (Required)):

The authors take a membrane-guided approach to identify cytoplasmic ribosomes oriented for protein import on the mitochondrial surface in *Saccharomyces cerevisiae* using cryo-electron tomography (cryo-ET). The authors show that ribosomes cluster, make multiple contacts with, and induce local changes to the mitochondrial membrane ultrastructure at import sites. Overall, this is a timely study that applies cryo-ET and new analytic methods developed by the authors to address an interesting question in mitochondrial protein import. The findings contradict the prevailing view that nuclear-encoded mitochondrial proteins are always imported post-translationally.

One of the more interesting observations is that at the putative co-translational import sites, the authors found ribosome clustering that coincides with striction of the OMM-IMM distance. The authors note that this would facilitate hand-off of protein import substrates from TOM to TIM complexes, but this should be discussed a little further. What are the cytosolic ribosomes actually doing to alter membrane geometry? Could it be that the nascent polypeptide itself is responsible for tethering the OMM-IMM and narrowing the IM space?

We agree with the reviewer that it is logical to assume that nascent polypeptides might be responsible for narrowing the OMM-IMM distance. However, existing data suggests that other factors may contribute to this remodeling. A previous study referenced in our response to Reviewer #1 (PMID: Table 3, PMID: 24942077) reported that quantum dot-labeled nascent chains associated with import supercomplexes did not alter the OMM-IMM distance. This finding implies that active co-translational import may specifically drive this effect. However, further experiments are needed to confirm this mechanism. While we envision our cryo-ET system being extremely valuable in testing these structure-function relationships in the future, we believe this topic falls beyond the scope of our current manuscript. We have now included some discussion of the results from this previous work in the revised draft of the paper, please see additional text in Reviewer 1 Comment #2.

The TOM complex docking in Fig. S4D is overly speculative and should be removed unless stronger evidence for the docking mode can be obtained.

Fig. S4D has been removed from the revised manuscript.

It was unclear why a GFP-TIM50 yeast strain was used. CLEM seems to have been used simply to identify yeast cells, nothing more. Has this strain been validated as non-perturbing for import?

We used this TIM50-GFP strain to verify through cryo-fluorescence microscopy that there were no obvious mitochondrial morphology defects in the cultures used for cryo-ET (representative images in response to Q1 of Reviewer #1), which show cryo-fluorescence images of mitochondrial network morphology. Most mitochondrial morphology of this strain is tubular, as shown in the image previously (Figure 3, PMID: 11978537). We have also included a revised **Supplementary Figure 1F** that shows no growth rate defects between the wild-type BY4741 and Tim50-GFP strain used in this study. Collectively, these indicate there is no significant perturbation of import.

How confident are the authors that no ER-bound ribosomes are being picked and incorporated into the sub-tomo average? These would look nearly identical based on the spherical mask chosen.

We are confident that no ER-bound ribosomes are being picked because the subset of ribosomes we included in the subtomogram average of co-translating ribosomes was selected based on the proximity of the ribosomal peptide exit tunnel to the nearest triangle within the outer mitochondrial membrane surface mesh reconstruction. We also performed a manual inspection in the visualization software ArtiaX (PMID: 36251681) of each of the 1,076 identified co-translating ribosomes and confirmed their localization near mitochondrial membranes (as opposed to other cellular membranes). Furthermore, we noted that there was only a single example of an ER membrane near mitochondria in our tomographic data (**Supplementary Figure 3D**), suggesting that co-translating ribosomes on mitochondria do not appear to be enriched at the ER-mitochondria membrane contact sites.

The authors have identified many ribosomes on the surface of the OMM that are not primed for import. What fraction of ribosomes are in the right orientation for import? What is the function of the ribosomes on the membrane in the wrong orientation? Could they just be nearby cytoplasmic ribosomes that aren't actually interacting with the OMM at all?

We thank the reviewer for this question. In our study, we used our morphometrics approach to specifically identify the population of ribosomes that is both (1) in close proximity to (0-250 Å) and (2) oriented with their peptide exit tunnel facing the outer mitochondrial membrane. If we only take into account proximity to membrane, the total number of ribosomes is approximately 2,823 in 87 CHX-treated datasets (87 tomograms). If we consider the relative orientation and only include the ribosomes with peptide tunnel facing the exit tunnel, we identify 1,076 ribosomes in CHX-treated datasets. This gives us two populations of ribosomes in close proximity to the outer mitochondrial membrane: those oriented for import (1,076 in CHX-treated datasets) and those not oriented for import (1,747 in CHX-treated datasets). This shows that ~38% of the ribosomes in close proximity to OMM are oriented for import upon CHX treatment. Given the high prevalence of ribosomes throughout the yeast cytoplasm, the remaining ribosomes might just be in close proximity to mitochondria due to random chance alone. Another alternative possibility is that these ribosomes in close proximity represent a pre-engagement state. Given the static nature of cryo-ET, it is difficult to imagine how we would test this hypothesis using our approach. However, we are excited that our work has the potential to inspire new hypotheses for us and others to test with complementary approaches. We have added some discussion of this in our revised text in **lines 91-101**:

“Considering distance alone, we identified 2,823 ribosomes in close proximity (0-250 Å) to the OMM. We reasoned that ribosomes engaged in mitochondrial protein import in cells would not only be near the OMM but would also likely adopt an orientation similar to that of co-translating ribosomes on the ER (Pfeffer, Burbaum et al. 2015, Gold, Chroscicki et al. 2017) (EMD-3764) and purified mitochondrial membranes (Gold, Chroscicki et al. 2017) (EMD-3762), with the peptide exit tunnel positioned within 95 Å from the membrane (Figure 1H). We, therefore, used this distance cut-off to identify the population of mitochondrially-associated cytoplasmic ribosomes likely engaged in co-translational protein import. This resulted in a total of 1,076 ribosomes optimally oriented for import. This suggests that, of the total ribosome population closest to the OMM, ~38% are likely engaged in co-translational import upon CHX treatment (Figure 1I, Supplementary Figure 1E). This analysis shows that the majority of ribosomes near the OMM are not optimally oriented for import and may either be in a pre-engagement state before import or simply located near the OMM due to random chance.”

December 17, 2024

RE: JCB Manuscript #202407110R

Danielle Grotjahn
Scripps Research Institute

Dear Dr. Grotjahn,

Thank you for submitting your revised manuscript entitled "Cytoplasmic ribosomes on mitochondria alter the local membrane environment for protein import." We would be happy to publish your paper in JCB pending final revisions necessary to meet our formatting guidelines (see details below).

A. MANUSCRIPT ORGANIZATION AND FORMATTING:

1) Text limits: Character count for Reports is < 20,000, not including spaces. Count includes title page, abstract, introduction, results, discussion, and acknowledgments. Count does not include materials and methods, figure legends, references, tables, or supplemental legends.

2) Figure formatting: Reports may have up to 5 main text figures. Scale bars must be present on all microscopy images, including inset magnifications. Please add a scale bar for Figure 3F.

Also, please avoid pairing red and green for images and graphs to ensure legibility for color-blind readers. If red and green are paired for images, please ensure that the particular red and green hues used in micrographs are distinctive with any of the colorblind types. If not, please modify colors accordingly or provide separate images of the individual channels.

3) Statistical analysis: Error bars on graphic representations of numerical data must be clearly described in the figure legend. The number of independent data points (n) represented in a graph must be indicated in the legend. Please, indicate whether 'n' refers to technical or biological replicates (i.e. number of analyzed cells, samples or animals, number of independent experiments). If independent experiments with multiple biological replicates have been performed, we recommend using distribution-reproducibility SuperPlots (please see Lord et al., JCB 2020) to better display the distribution of the entire dataset, and report statistics (such as means, error bars, and P values) that address the reproducibility of the findings.

Statistical methods should be explained in full in the materials and methods. For figures presenting pooled data the statistical measure should be defined in the figure legends. Please also be sure to indicate the statistical tests used in each of your experiments (both in the figure legend itself and in a separate methods section) as well as the parameters of the test (for example, if you ran a t-test, please indicate if it was one- or two-sided, etc.). Also, if you used parametric tests, please indicate if the data distribution was tested for normality (and if so, how). If not, you must state something to the effect that "Data distribution was assumed to be normal but this was not formally tested."

4) Materials and methods: Should be comprehensive and not simply reference a previous publication for details on how an experiment was performed. Please provide full descriptions (at least in brief) in the text for readers who may not have access to referenced manuscripts. The text should not refer to methods "...as previously described."

5) For all cell lines, vectors, constructs/cDNAs, etc. - all genetic material: please include database / vendor ID (e.g. Addgene, ATCC, etc.) or if unavailable, please briefly describe their basic genetic features, even if described in other published work or gifted to you by other investigators (and provide references where appropriate). Please be sure to provide the sequences for all of your oligos: primers, si/shRNA, RNAi, gRNAs, etc. in the materials and methods. You must also indicate in the methods the source, species, and catalog numbers/vendor identifiers (where appropriate) for all of your antibodies, including secondary. If antibodies are not commercial, please add a reference citation if possible.

6) Microscope image acquisition: The following information must be provided about the acquisition and processing of images:

- Make and model of microscope
- Type, magnification, and numerical aperture of the objective lenses
- Temperature
- Imaging medium

- e. Fluorochromes
- f. Camera make and model
- g. Acquisition software
- h. Any software used for image processing subsequent to data acquisition. Please include details and types of operations involved (e.g., type of deconvolution, 3D reconstitutions, surface or volume rendering, gamma adjustments, etc.).

7) References: There is no limit to the number of references cited in a manuscript. References should be cited parenthetically in the text by author and year of publication. Abbreviate the names of journals according to PubMed.

8) Supplemental materials: Reports may have up to 5 supplemental figures and 10 videos. Please also note that tables, like figures, should be provided as individual, editable files. A summary of all supplemental material should appear at the end of the Materials and methods section. Please include one brief sentence per item.

9) eTOC summary: A ~40-50 word summary that describes the context and significance of the findings for a general readership should be included on the title page. The statement should be written in the present tense and refer to the work in the third person. It should begin with "First author name(s) et al..." to match our preferred style.

10) Conflict of interest statement: JCB requires inclusion of a statement in the acknowledgements regarding competing financial interests. If no competing financial interests exist, please include the following statement: "The authors declare no competing financial interests." If competing interests are declared, please follow your statement of these competing interests with the following statement: "The authors declare no further competing financial interests."

11) A separate author contribution section is required following the Acknowledgments in all research manuscripts. All authors should be mentioned and designated by their first and middle initials and full surnames. We encourage use of the CRediT nomenclature (<https://casrai.org/credit/>).

12) ORCID IDs: ORCID IDs are unique identifiers allowing researchers to create a record of their various scholarly contributions in a single place. Please note that ORCID IDs are required for all authors. At resubmission of your final files, please be sure to provide your ORCID ID and those of all co-authors.

13) Journal of Cell Biology now requires a data availability statement for all research article submissions. These statements will be published in the article directly above the Acknowledgments. The statement should address all data underlying the research presented in the manuscript. Please visit the JCB instructions for authors for guidelines and examples of statements at (<https://rupress.org/jcb/pages/editorial-policies#data-availability-statement>).

B. FINAL FILES:

Thank you for your attention to these final processing requirements. Please revise and format the manuscript and upload

materials within 7 days. If you need an extension for whatever reason, please let us know and we can work with you to determine a suitable revision period.

Thank you for this interesting contribution, we look forward to publishing your paper in Journal of Cell Biology.

Sincerely,

Jodi Nunnari, PhD
Editor-in-Chief
Journal of Cell Biology

Dan Simon, PhD
Scientific Editor
Journal of Cell Biology

Reviewer #1 (Comments to the Authors (Required)):

The authors addressed adequately the points raised on the previous submission. Even though some aspects will have to be clarified in the future, this study is interesting and well suited for the JCB in its present form. I therefore support its publication.

Reviewer #3 (Comments to the Authors (Required)):

The authors have done a nice job of responding to the reviewer comments and I am happy to recommend publication in the present form.